# Beyond Model Ranking: Predictability-Aligned Evaluation for Time Series Forecasting

**Wanjin Feng** [1]  **Yuan Yuan** [1]  **Jingtao Ding** [1]  **Yong Li** [1]

## Abstract

In the era of increasingly complex AI models for time series forecasting, progress is often measured by marginal improvements on benchmark leaderboards. However, standard evaluations rely on aggregate metrics (e.g., MSE) that conflate model capability with the intrinsic difficulty of the evaluated instances. To address this, we propose a diagnostic framework anchored in **Spectral Coherence Predictability (SCP)**, which provides an efficient $\mathcal{O}(N \log N)$ per-instance difficulty reference and yields a corresponding linear MSE lower bound. Complementing this, we introduce the **Linear Utilization Ratio (LUR)** to quantify how effectively models exploit linearly predictable structures across frequencies. Experiments on synthetic and real-world benchmarks show that SCP aligns strongly with realized forecasting errors across diverse state-of-the-art forecasters. Using this lens, we uncover **"predictability drift,"** revealing that task difficulty is not static but fluctuates significantly over time and variables. Furthermore, stratified evaluation exposes complementary architectural strengths across distinct frequency bands and difficulty regimes. Overall, we advocate moving beyond leaderboard-style ranking toward a more insightful, predictability-aware evaluation that fosters fairer model comparisons and a deeper understanding of model behavior. Code and data are available at https://github.com/WanjinVon/TS_Predictability.

## 1. Introduction

Despite the proliferation of ever-more-complex models for time-series forecasting, true progress in the field remains notoriously difficult to measure (Bergmeir, 2024). The community relies on standard metrics, such as Mean Squared Error (MSE) and Mean Absolute Error (MAE), which summarize prediction errors but provide little insight into why those errors occur. This is problematic because aggregate errors conflate model limitations with instance-level predictability of the data, which changes across time, channels, and frequency bands. This ambiguity leads to an evaluation dilemma: a sophisticated model may appear inferior to a baseline simply because the test sequence is regular and therefore easy to predict. Consequently, these metrics obscure the origins of performance gaps and hinder scientific iteration. Beyond mere ranking, the field requires a diagnostic framework that quantifies instance difficulty in alignment with forecasting objectives, enabling stratified evaluation and revealing where models under-utilize available information (Erkintalo, 2015).

To resolve this evaluation dilemma, we must quantify time-series predictability to establish a difficulty reference for each forecasting instance. However, designing such a difficulty metric for modern deep-learning forecasting presents several challenges (Pennekamp et al., 2019). First, the metric must be task-aligned: its theoretical foundation should cohere with multi-horizon forecasting under a squared-error loss, rather than traditional single-step classification accuracy (Mishra & Palanisamy, 2018). Second, it must be computationally efficient to handle the massive, high-dimensional time series prevalent today (Fiecas et al., 2019). Finally, a single global predictability score is insufficient: a practical tool must be diagnostic, offering insights to reveal where a model succeeds or fails in capturing predictable patterns.

Viewed through the lens of these challenges, existing tools are ill-suited for this purpose. Traditional proxies for predictability, such as entropy-rate estimators and Lempel-Ziv complexity, suffer from a fundamental paradigm mismatch (Aboy et al., 2006). They were primarily developed for symbolic dynamics and discrete prediction settings, where the goal is to characterize sequence complexity or next-

---

[1]Department of Electronic Engineering, Tsinghua University, Beijing, China.. Correspondence to: Yuan Yuan <y-yuan20@tsinghua.org.cn>, Yong Li <liyong07@tsinghua.edu.cn>.

*Proceedings of the 43rd International Conference on Machine Learning*, Seoul, South Korea. PMLR 306, 2026. Copyright 2026 by the author(s).

symbol predictability under 0–1 loss, rather than multi-horizon regression performance under squared error (Zhao et al., 2021). Computationally, they are often prohibitively expensive—typically entailing quadratic-to-cubic complexity—and rely on strict stationarity assumptions, rendering them impractical for the large-scale, non-stationary datasets common in modern applications (Kontoyiannis et al., 2002; Wyner & Ziv, 2002). Finally, these approaches typically yield a single global score, offering limited diagnostic insight into where difficulty arises or how a model fails to exploit available information across time, channels, or frequency bands. These gaps motivate a new, forecasting-oriented framework for quantifying instance difficulty and diagnosing model–data mismatch.

To bridge this gap, we introduce a diagnostic framework grounded in spectral coherence that is computationally efficient, aligned with the squared-error forecasting objective, and designed to provide multi-scale insight. Our framework consists of two core components: 1) **Spectral Coherence Predictability (SCP)**, a per-instance difficulty reference that quantifies the amount of linearly exploitable information available for forecasting. SCP can be computed in $O(N \log N)$ time and supports scalable, instance-level stratification. 2) **Linear Utilization Ratio (LUR)**, a frequency-resolved diagnostic that quantifies how effectively a model exploits linearly predictable component across different spectral bands, enabling fine-grained assessments of under-utilization, saturation, and potential gains from non-linear modeling. Together, these tools shift evaluation from simple model ranking toward model–data diagnostics, enabling difficulty-aware comparisons and actionable insights into when and where models fail to exploit available structure. Across synthetic and real-world benchmarks, we show that SCP is well-calibrated as an instance-difficulty proxy and strongly correlates with the empirical errors of state-of-the-art forecasters. Moreover, the proposed diagnostics reveal substantial time variation in instance difficulty (predictability drift), enabling fairer stratified evaluation that uncovers architecture-dependent strengths beyond what aggregate scores can capture, and providing practical guidance for developing more robust forecasting models.

In summary, our contributions are as follows:

- We systematically address evaluation ambiguity in modern time-series forecasting by introducing a predictability-aware diagnostic framework that separates model performance from instance difficulty.

- We propose Spectral Coherence Predictability (SCP), a computationally efficient and task-aligned instance-difficulty reference, together with Linear Utilization Ratio (LUR), a frequency-resolved diagnostic for analyzing how models utilize linearly predictable components.

- Extensive experiments validates this framework's alignment with state-of-the-art models. We leverage it to uncover predictability drift and to enable stratified evaluation that highlights complementary strengths across different models.

## 2. Related Work

Our work is positioned at the intersection of two key research areas: the quantification of sequence predictability and the use of spectral methods for time-series analysis.

**Predictability of Time Series.** Entropy-based notions have long been used to proxy sequence predictability, from Shannon's entropy and entropy rate to variants usable on continuous data (approximate, sample, fuzzy, and permutation entropy) (Shannon, 1948; Pincus, 1991; Richman & Moorman, 2000; Bandt & Pompe, 2002; Garland et al., 2014). Compression-driven estimators (e.g., Lempel–Ziv) provide nonparametric estimates of entropy rate for symbolic, stationary sources (Ziv & Lempel, 1977). These approaches have also been popular in human mobility, where spatio-temporal regularity supports predictability limits under coarse symbolizations (González et al., 2008; Song et al., 2010; Wang et al., 2021). However, they face three key limitations for general forecasting: (i) computational burden and the need for discretization of continuous data; (ii) theoretical misalignment with multi-step squared-loss objectives; and (iii) sensitivity of differential entropy to reparameterization and divergence issues in non-stationary settings (Mohammed et al., 2024). Consequently, existing predictability studies have primarily focused on intrinsic predictability or theoretical performance limits (Song et al., 2010; Chen et al., 2022; Mohammed et al., 2024), but are not directly designed for real-valued multi-step time-series forecasting, and have rarely been framed as a predictability-centered evaluation framework for understanding realized forecasting performance, model bias across regimes, and temporal predictability drift.

**Spectral Analysis in Time Series Forecasting.** Spectral analysis is a cornerstone of time-series modeling, inspiring many recent deep learning architectures. For instance, Autoformer was designed with an auto-correlation mechanism to discover period-based dependencies efficiently (Wu et al., 2021). FEDformer directly integrates Fourier transforms into attention for frequency-domain computation with reduced complexity (Zhou et al., 2022). TimesNet captures complex multi-periodicity by transforming the 1D time series into a 2D representation for analysis (Wu et al., 2023). These methods all leverage spectral properties to build better models. In contrast, our work uses spectral coherence to build a novel diagnostic framework for analyzing data predictability and evaluating the utilization of existing models.

## 3. Preliminaries

**Problem setup and notation.** We focus on a setting in which an observed sequence is decomposed into a past (history) portion used as input and a future portion serving as ground-truth for evaluation. Formally, a sample consists of a history $\boldsymbol{x} \in \mathbb{R}^{N_x}$ and a future $\boldsymbol{y} \in \mathbb{R}^{N_y}$ drawn from a distribution $\mathbb{D}$. The goal is to learn a measurable predictor $f : \mathbb{R}^{N_x} \to \mathbb{R}^{N_y}$ that produces a forecast $\widehat{\boldsymbol{y}} = f(\boldsymbol{x})$. We evaluate predictions with the mean squared error (MSE) per forecast step:

$$\mathrm{MSE}(f; \boldsymbol{x}, \boldsymbol{y}) \;=\; \frac{1}{N_y} \|f(\boldsymbol{x}) - \boldsymbol{y}\|_2^2, \qquad (1)$$

where $\| \cdot \|_2$ denotes the Euclidean norm. The objective is to minimize the expected risk $\mathbb{E}_{(\boldsymbol{x}, \boldsymbol{y}) \sim \mathcal{D}}[\mathrm{MSE}(f; \boldsymbol{x}, \boldsymbol{y})]$.

**Intrinsic predictability via Bayes risk.** Under the MSE metric, the risk-minimizing predictor is the conditional expectation $f^\star(\boldsymbol{x}) = \mathbb{E}[\boldsymbol{y} \mid \boldsymbol{x}]$ (Chen et al., 2016). The corresponding minimum achievable risk (Bayes risk) is

$$\mathrm{MSE}^\star \;=\; \mathbb{E}\left[ \frac{1}{N_y} \big\| \boldsymbol{y} - \mathbb{E}[\boldsymbol{y} \mid \boldsymbol{x}] \big\|_2^2 \right]. \qquad (2)$$

Using the unconditional variance $\mathrm{Var}(\boldsymbol{y})$ as a baseline, we define intrinsic predictability as the normalized reduction of uncertainty:

$$\mathcal{P}_{xy}^\star \;=\; 1 - \frac{\mathrm{MSE}^\star}{\mathrm{Var}(\boldsymbol{y})}. \qquad (3)$$

By the law of total variance, $\mathrm{Var}(\boldsymbol{y}) = \mathrm{MSE}^\star + \mathrm{Var}\big(\mathbb{E}[\boldsymbol{y} \mid \boldsymbol{x}]\big)$, hence $\mathcal{P}_{xy}^\star \in [0, 1]$. At the extremes, $\mathcal{P}_{xy}^\star = 1$ if and only if $\mathrm{Var}(\boldsymbol{y} \mid \boldsymbol{x}) = 0$ almost surely, i.e., $\boldsymbol{y}$ is a deterministic function of $\boldsymbol{x}$. Conversely, $\mathcal{P}_{xy}^\star = 0$ if and only if $\mathrm{Var}\big(\mathbb{E}[\boldsymbol{y} \mid \boldsymbol{x}]\big) = 0$, i.e., the conditional mean $\mathbb{E}[\boldsymbol{y} \mid \boldsymbol{x}]$ is constant and $\boldsymbol{x}$ conveys no information for predicting the mean of $\boldsymbol{y}$.

While $\mathcal{P}_{xy}^\star$ provides a rigorous theoretical ceiling, it remains computationally elusive. The conditional distribution $\mathbb{P}(\boldsymbol{y}|\boldsymbol{x})$ is inaccessible for high-dimensional time series given finite data and unknown generative processes. This raises a critical practical question: Can we define a computable surrogate for difficulty that is computationally efficient and remains strongly correlated with real-world model performance?

## 4. Method

### 4.1. Spectral Coherence Predictability

To answer this question, we introduce the Spectral Coherence Predictability (SCP). Bridging the gap between the theoretical Bayes risk (Eq. 3) and practical computation, SCP serves as a tractable surrogate. Instead of estimating the full

---

**Algorithm 1** Spectral Coherence Predictability (SCP)

**Require:** History $\boldsymbol{x} \in \mathbb{R}^{N_x}$, future $\boldsymbol{y} \in \mathbb{R}^{N_y}$; Welch parameters; optional frequency band $\mathcal{F}_b$.
**Ensure:** MSE linear lower bound $\mathrm{MSE}_{\mathrm{lb}}$ and predictability $\mathcal{P}_{xy}$.

1: **Mean removal:** $m_x \leftarrow \mathrm{mean}(x), \; m_y \leftarrow \mathrm{mean}(y)$; $\Delta^2 \leftarrow (m_y - m_x)^2; \; x \leftarrow x - m_x, \; y \leftarrow y - m_y$.
2: **Welch spectra:** Compute the PSD $\widehat{S}_{xx}(f), \widehat{S}_{yy}(f)$ and the CPSD $\widehat{S}_{xy}(f)$ on the discrete frequency domain $\mathcal{F}$.
3: **Squared coherence:**

$$\gamma^2(f) \;\leftarrow\; \frac{|\widehat{S}_{xy}(f)|^2}{\big(\widehat{S}_{xx}(f) + \varepsilon\big)\big(\widehat{S}_{yy}(f) + \varepsilon\big)} \;\in [0, 1].$$

4: **Residual spectrum:** $\widehat{S}_e(f) \leftarrow \widehat{S}_{yy}(f)\big(1 - \gamma^2(f)\big)$, $\forall f \in \mathcal{F}$.
5: **Frequency set:** $\mathcal{F}_\star \leftarrow \mathcal{F}_b$ if a band $\mathcal{F}_b$ is provided; otherwise $\mathcal{F}_\star \leftarrow \mathcal{F}$.
6: **Aggregate:**

$$\widehat{\mathrm{Var}}(y) \;\leftarrow\; \sum_{f \in \mathcal{F}_\star} \widehat{S}_{yy}(f), \mathrm{MSE}_{\mathrm{lb}} \;\leftarrow\; \Delta^2 + \sum_{f \in \mathcal{F}_\star} \widehat{S}_e(f).$$

7: **Predictability:** $\mathcal{P}_{xy} \leftarrow 1 - \mathrm{MSE}_{\mathrm{lb}} / \widehat{\mathrm{Var}}(y)$.
8: **Return:** $\mathrm{MSE}_{\mathrm{lb}}, \; \mathcal{P}_{xy}$.

---

conditional distribution, SCP leverages frequency-domain structure to quantify how much of the future segment $\boldsymbol{y}$ is linearly explainable by the history $\boldsymbol{x}$.

We operate in the frequency domain using Welch's method. Let $\widehat{S}_{yy}(f)$ and $\widehat{S}_{xx}(f)$ denote the power spectral densities (PSD) of $\boldsymbol{y}$ and $\boldsymbol{x}$, and let $\widehat{S}_{xy}(f)$ denote their cross–power spectral density (CPSD). All spectra are computed on the same discrete Fourier transform (DFT) grid with identical Welch parameters after mean removal. The squared coherence between $\boldsymbol{y}$ and $\boldsymbol{x}$ is

$$\gamma_{xy}^2(f) \;=\; \frac{\big|\widehat{S}_{xy}(f)\big|^2}{\big(\widehat{S}_{xx}(f) + \varepsilon\big)\big(\widehat{S}_{yy}(f) + \varepsilon\big)} \in [0, 1], \quad (4)$$

where $\varepsilon > 0$ is a small term for numerical stability (Mandel & Wolf, 1976; Wang et al., 2019). Interpreting $\gamma_{xy}^2(f)$ as a linearly explained–power ratio, the unexplained (residual) spectrum is

$$\widehat{S}_e(f) \;=\; \widehat{S}_{yy}(f)\big(1 - \gamma_{xy}^2(f)\big). \qquad (5)$$

Let $\mathcal{F}$ denote the discrete frequency domain under our normalization, so that the total spectral power equals the sample variance, i.e.,

$$\widehat{\mathrm{Var}}(\boldsymbol{y}) = \sum_{f \in \mathcal{F}} \widehat{S}_{yy}(f). \qquad (6)$$

After mean removal, the residual spectral power $\sum_{f \in \mathcal{F}} \widehat{S}_e(f)$ gives a lower bound on the MSE of any linear time-invariant predictor, and thus serves as a stationary-linear reference for instance-level difficulty (Davenport Jr et al., 1958). However, the means of the history and prediction windows may differ. To conservatively account for such boundary mean mismatch, we add a mean-shift term

$$\Delta^2 \;=\; \big(\mathrm{mean}(\boldsymbol{y}) - \mathrm{mean}(\boldsymbol{x})\big)^2.$$

This gives the following conservative linear reference error:

$$\mathrm{MSE}_{\mathrm{lb}} \;=\; \Delta^2 \;+\; \sum_{f \in \mathcal{F}} \widehat{S}_e(f). \tag{7}$$

Here, $\mathrm{MSE}_{\mathrm{lb}}$ should be interpreted as a conservative surrogate lower bound relative to the chosen stationary-linear reference. The SCP estimate of predictability is then defined as

$$\mathcal{P}_{xy} \;=\; \max\left\{0,\, 1 - \frac{\mathrm{MSE}_{\mathrm{lb}}}{\widehat{\mathrm{Var}}(\boldsymbol{y})}\right\} \in [0,1]. \tag{8}$$

Algorithm 1 summarizes the steps. Computationally, with fast Fourier transform, SCP costs $\mathcal{O}(N \log N)$ per sample. This is substantially lower than matching–based Lempel–Ziv–style predictability estimators, which typically entail at least quadratic–to–cubic time in sequence length (e.g., $\mathcal{O}(N^3)$ in naive implementations) and usually target single–step predictability, whereas SCP yields a multi–step estimate aligned with the evaluation horizon.

**Theoretical interpretation.** If $(\boldsymbol{x}, \boldsymbol{y})$ is jointly Gaussian and wide–sense stationary around the boundary, the Bayes predictor is linear (Ko & Fox, 2009). In this case, Eq. (8) is a consistent estimator of the intrinsic predictability $\mathcal{P}_{xy}^{\star}$ as the effective sample size grows. For general processes, highly non-linear or non-stationary components often manifest as stochasticity (noise) in limited-sample regimes. Therefore, by treating these components as unexplained variance, SCP provides a robust and conservative baseline. It captures the reliable signal structure while avoiding the pitfall of overfitting to chaotic dynamics that are theoretically deterministic but practically unpredictable. Nevertheless, our framework is not limited to the linear setting. We discuss extensions that incorporate nonlinear dependencies in Appendix B.2.

### 4.2. Linear Utilization Ratio

Instead of relying solely on pointwise error metrics (e.g., MSE/MAE), which summarize how close a forecast is to the target but not why it succeeds or fails, we introduce a frequency-resolved diagnostic.

Our method, detailed in Algorithm 2, is built on two key quantities: The first is the history–future coherence, $\gamma_{yx}^2(f)$ (Eq. (4)), which quantifies the fraction of the target power

---

**Algorithm 2** Linear Utilization Ratio (LUR)

**Require:** History $\boldsymbol{x} \in \mathbb{R}^{N_x}$, future $\boldsymbol{y} \in \mathbb{R}^{N_y}$, model prediction $\hat{\boldsymbol{y}} \in \mathbb{R}^{N_y}$; Welch parameters; optional frequency band $\mathcal{F}_b$.

**Ensure:** Model–explained power $P_{\mathrm{model}}$; linear utilization ratio LUR.

1: **Mean removal:** $\boldsymbol{x} \leftarrow \boldsymbol{x} - \mathrm{mean}(\boldsymbol{x})$; $\boldsymbol{y} \leftarrow \boldsymbol{y} - \mathrm{mean}(\boldsymbol{y})$; $\hat{\boldsymbol{y}} \leftarrow \hat{\boldsymbol{y}} - \mathrm{mean}(\hat{\boldsymbol{y}})$.

2: **Welch spectra:**
$$\widehat{S}_{xx}(f),\ \widehat{S}_{yy}(f),\ \widehat{S}_{\hat{y}\hat{y}}(f),\ \widehat{S}_{xy}(f),\ \widehat{S}_{y\hat{y}}(f),\quad \forall f \in \mathcal{F}.$$

3: **Coherences:**
$$\gamma_{yx}^2(f) \leftarrow \frac{|\widehat{S}_{yx}(f)|^2}{\big(\widehat{S}_{yy}(f) + \varepsilon\big)\big(\widehat{S}_{xx}(f) + \varepsilon\big)},$$

$$\gamma_{y\hat{y}}^2(f) \leftarrow \frac{|\widehat{S}_{y\hat{y}}(f)|^2}{\big(\widehat{S}_{yy}(f) + \varepsilon\big)\big(\widehat{S}_{\hat{y}\hat{y}}(f) + \varepsilon\big)}.$$

4: **Frequency set:** $\mathcal{F}_{\star} \leftarrow \mathcal{F}_b$ if a band $\mathcal{F}_b$ is provided; otherwise $\mathcal{F}_{\star} \leftarrow \mathcal{F}$.

5: **Power–weighted aggregation:**
$$P_{\mathrm{model}} \leftarrow \sum_{f \in \mathcal{F}_{\star}} \gamma_{y\hat{y}}^2(f)\, \widehat{S}_{yy}(f),$$

$$P_{\mathrm{linear}} \leftarrow \sum_{f \in \mathcal{F}_{\star}} \gamma_{yx}^2(f)\, \widehat{S}_{yy}(f).$$

6: **LUR ratio:** $\mathrm{LUR} \leftarrow P_{\mathrm{model}}/P_{\mathrm{linear}}$.

7: **Return:** $P_{\mathrm{model}}$, $P_{\mathrm{linear}}$, LUR.

---

at frequency $f$ that is linearly associated with the history $\boldsymbol{x}$. The second is the prediction–target coherence, measuring how much of $\boldsymbol{y}$'s power is captured by the model prediction $\hat{\boldsymbol{y}}$ at frequency $f$:

$$\gamma_{y\hat{y}}^2(f) \;=\; \frac{\big|\widehat{S}_{y\hat{y}}(f)\big|^2}{\big(\widehat{S}_{yy}(f) + \varepsilon\big)\big(\widehat{S}_{\hat{y}\hat{y}}(f) + \varepsilon\big)} \;\in\; [0,1]. \tag{9}$$

Comparing these two coherences yields a per-frequency diagnosis:

- **Under-utilization** ($\gamma_{y\hat{y}}^2 < \gamma_{yx}^2$): The model fails to capture simple linear correlations present in the history, indicating optimization failure or underfitting.

- **Saturation** ($\gamma_{y\hat{y}}^2 \approx \gamma_{yx}^2$): The model has fully exhausted the linear information in $\boldsymbol{x}$, hitting the baseline performance ceiling.

- **Non-linear Advantage** ($\gamma_{y\hat{y}}^2 > \gamma_{yx}^2$): The model surpasses the instance-wise linear limit. This indicates the

successful exploitation of non-linear dynamics or global inductive biases learned from the training set (cross-instance patterns).

To summarize over frequencies while emphasizing high-energy regions, we compute power-weighted aggregates on the discrete frequency domain $\mathcal{F}$:

$$P_{\text{model}} = \sum_{f \in \mathcal{F}} \gamma_{y\hat{y}}^2(f)\, \widehat{S}_{yy}(f), \qquad (10)$$

$$P_{\text{linear}} = \sum_{f \in \mathcal{F}} \gamma_{yx}^2(f)\, \widehat{S}_{yy}(f). \qquad (11)$$

To explicitly quantify the efficiency with which a model captures this predictable energy, we define the Linear Utilization Ratio (LUR):

$$\text{LUR} \;=\; \frac{P_{\text{model}}}{P_{\text{linear}}} \;\geq\; 0. \qquad (12)$$

An $\text{LUR} < 1$ indicates information loss, $\text{LUR} \approx 1$ suggests that the model approaches the linear optimum, whereas $\text{LUR} > 1$ indicates additional predictive gains enabled by cross-channel linear dependencies, nonlinear structures, or global modeling capabilities.

To analyze behavior across scales, we additionally partition the discrete frequency domain into disjoint bands $\{\mathcal{F}_b\}_{b=1}^{B}$ (e.g., low/mid/high), using the band partition as in Algorithms 1 and 2. This yields band-limited counterparts $\text{MSE}_{\text{lb},b}$, $\mathcal{P}_{xy,b}$, and $\text{LUR}_b$, which enable localized diagnosis of under-use, saturation, or beyond-linear gains within each frequency band.

# 5. Experiments

This section empirically evaluates our framework across both controlled synthetic environments and extensive real-world benchmarks. Detailed experimental settings are provided in the appendix. Our analysis is guided by three research questions:

- **Q1 (Calibration):** Is the proposed SCP metric well-calibrated as an instance-predictability proxy? Specifically, does it correlate with empirical forecasting errors in practice, even for complex non-linear models evaluated on real-world data (Secs. 5.1 and 5.2)?

- **Q2 (Dynamics):** What insights can this difficulty-aware lens reveal about time-varying data characteristics, such as predictability drift (Sec. 5.3)?

- **Q3 (Diagnostics):** How can the framework facilitate a more comprehensive, stratified evaluation to uncover the differential strengths of forecasting architectures? (Secs. 5.4 and 5.5)

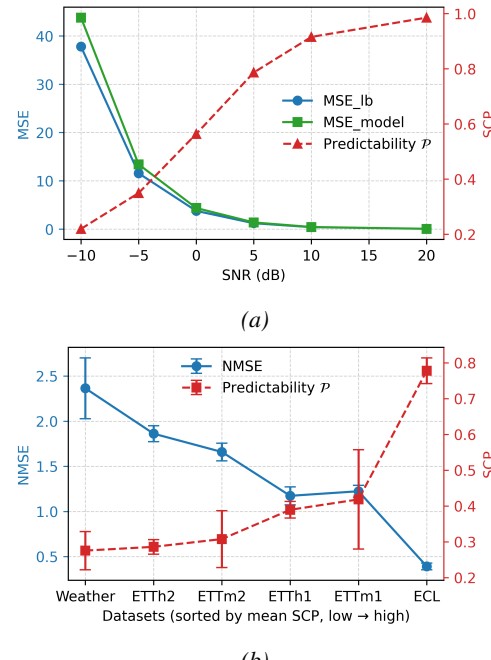

*(a)*

*(b)*

*Figure 1.* Calibration of SCP against Model Error. (a) Synthetic Validation: MSE of the best linear predictor on a synthetic Gaussian process with varying SNR. (b) Real-World Alignment: Average performance of state-of-the-art prediction models on real datasets. We report normalized MSE (NMSE), obtained by dividing MSE by the corresponding variance.

## 5.1. Toy Study

We first validate our proposed SCP score $\mathcal{P}$ and its associated linear reference error $\text{MSE}_{\text{lb}}$ in a controlled synthetic setting. Specifically, we consider a Gaussian process with additive noise at varying Signal-to-Noise Ratios (SNRs) and evaluate an optimal linear forecaster (Fig. 1a). As noise decreases (higher SNR), $\mathcal{P}$ increases monotonically toward one, while the model MSE approaches $\text{MSE}_{\text{lb}}$. Across all SNRs, $\text{MSE}_{\text{lb}}$ remains below the realized MSE, and the gap shrinks at high SNR, indicating that the linear forecaster increasingly saturates the data-implied linear reference in the near noise-free regime. Overall, this toy study supports both the *calibration* of our metric (monotonic response to controllable noise) and the *tightness* of the reference in the linear regime.

## 5.2. Aligning Predictability and Forecasting Performance

Having validated SCP in a controlled synthetic setting, we next examine whether it aligns with forecasting performance on real-world benchmarks. We evaluate five state-of-the-art (SOTA) models, including Transformer-based methods (iTransformer, PatchTST), a CNN-based model (TimesNet), and linear baselines (DLinear, TimeMixer), across widely

*Table 1.* Long-term multivariate forecasting results. We report MSE, MAE, NMSE for forecasting lengths equal to history length $N \in \{96, 192, 336, 720\}$ under an identical protocol (same preprocessing and no drop-last). **Bold** marks the best (lowest) MSE/MAE per column across models. *Average* rows give the column-wise mean across models. Predictability reports the per-task linear MSE lower bound ($MSE_{lb}$) and SCP $\mathcal{P}$ (higher is easier). Results on additional datasets are provided in Sec. C.3.

| Models | Metric | ETTh1 | | | | ETTh2 | | | | ETTm1 | | | | ETTm2 | | | | ECL | | | | Weather | | | |
|---|---|---|---|---|---|---|---|---|---|---|---|---|---|---|---|---|---|---|---|---|---|---|---|---|---|
| | | 96 | 192 | 336 | 720 | 96 | 192 | 336 | 720 | 96 | 192 | 336 | 720 | 96 | 192 | 336 | 720 | 96 | 192 | 336 | 720 | 96 | 192 | 336 | 720 |
| iTransformer | MSE | 0.387 | 0.441 | 0.471 | 0.700 | 0.301 | 0.381 | 0.426 | 0.425 | 0.342 | 0.345 | 0.379 | 0.448 | 0.186 | 0.254 | 0.289 | 0.382 | **0.148** | 0.156 | 0.170 | **0.194** | 0.176 | 0.214 | 0.255 | 0.353 |
| | MAE | 0.405 | 0.440 | 0.464 | 0.608 | 0.350 | 0.405 | 0.438 | 0.455 | 0.377 | 0.378 | 0.403 | 0.449 | 0.272 | 0.319 | 0.341 | 0.407 | **0.239** | 0.250 | 0.266 | **0.287** | 0.216 | 0.255 | 0.290 | 0.357 |
| (Liu et al., 2024) | NMSE | 1.121 | 1.095 | 1.056 | 1.292 | 1.598 | 1.829 | 1.563 | 1.432 | 1.340 | 1.167 | 1.140 | 1.159 | 1.609 | 1.696 | 1.595 | 1.917 | 0.288 | 0.268 | 0.280 | 0.319 | 2.629 | 2.353 | 2.032 | 2.150 |
| | R | 0.844 | 0.876 | 0.899 | 0.747 | 0.907 | 0.883 | 0.877 | 0.842 | 0.845 | 0.782 | 0.803 | 0.869 | 0.868 | 0.834 | 0.898 | 0.840 | 0.723 | 0.778 | 0.821 | 0.826 | 0.900 | 0.876 | 0.801 | 0.824 |
| TimeMixer | MSE | **0.381** | 0.440 | 0.482 | 0.631 | **0.289** | 0.377 | 0.390 | 0.435 | **0.322** | 0.337 | 0.380 | 0.484 | **0.176** | 0.231 | 0.280 | 0.376 | 0.153 | 0.155 | 0.172 | 0.214 | **0.169** | **0.198** | 0.249 | 0.347 |
| | MAE | 0.400 | 0.434 | 0.460 | 0.561 | **0.340** | 0.406 | 0.423 | 0.458 | 0.359 | 0.372 | 0.396 | 0.469 | 0.259 | 0.296 | 0.332 | 0.390 | 0.245 | 0.244 | 0.264 | 0.310 | **0.215** | **0.242** | 0.291 | 0.355 |
| (Wang et al., 2024) | NMSE | 1.131 | 1.069 | 1.062 | 1.155 | 1.540 | 1.724 | 1.524 | 1.446 | 1.303 | 1.105 | 1.135 | 1.202 | 1.493 | 1.485 | 1.498 | 1.689 | 0.282 | 0.274 | 0.286 | 0.334 | 2.602 | 2.161 | 2.107 | 2.162 |
| | R | 0.815 | 0.889 | 0.848 | 0.793 | 0.916 | 0.801 | 0.909 | 0.906 | 0.829 | 0.781 | 0.752 | 0.798 | 0.867 | 0.910 | 0.852 | 0.887 | 0.706 | 0.607 | 0.682 | 0.887 | 0.911 | 0.862 | 0.885 | 0.852 |
| DLinear | MSE | 0.383 | **0.422** | 0.447 | 0.507 | 0.329 | 0.375 | 0.463 | 0.740 | 0.346 | 0.342 | 0.372 | **0.415** | 0.187 | 0.242 | 0.278 | 0.374 | 0.195 | 0.163 | 0.169 | 0.197 | 0.197 | 0.225 | 0.263 | 0.315 |
| | MAE | **0.396** | **0.421** | 0.448 | 0.517 | 0.380 | 0.410 | 0.472 | 0.609 | 0.374 | 0.369 | **0.389** | **0.415** | 0.281 | 0.315 | 0.338 | 0.406 | 0.277 | 0.259 | 0.268 | 0.295 | 0.255 | 0.282 | 0.314 | 0.354 |
| (Zeng et al., 2023) | NMSE | 1.214 | 1.143 | 1.310 | 1.782 | 2.927 | 2.067 | 2.728 | 3.896 | 1.327 | 1.205 | 1.208 | 1.121 | 1.676 | 1.722 | 1.620 | 1.793 | 0.868 | 0.678 | 0.574 | 0.684 | 3.507 | 2.899 | 2.474 | 2.069 |
| | R | 0.869 | 0.878 | 0.872 | 0.804 | 0.845 | 0.880 | 0.798 | 0.439 | 0.868 | 0.887 | 0.819 | 0.884 | 0.833 | 0.813 | 0.910 | 0.902 | 0.880 | 0.867 | 0.909 | 0.864 | 0.924 | 0.931 | 0.911 | 0.923 |
| PatchTST | MSE | 0.391 | 0.429 | **0.436** | **0.465** | 0.293 | **0.357** | **0.363** | **0.406** | **0.322** | **0.328** | **0.365** | 0.417 | 0.177 | **0.230** | **0.276** | **0.356** | 0.167 | **0.151** | **0.167** | 0.212 | 0.176 | 0.202 | **0.247** | **0.309** |
| | MAE | 0.403 | 0.426 | **0.440** | **0.482** | 0.342 | **0.387** | 0.390 | **0.442** | **0.358** | **0.364** | 0.390 | 0.419 | 0.252 | **0.294** | **0.329** | **0.385** | 0.252 | **0.242** | **0.258** | 0.304 | 0.217 | 0.243 | **0.281** | **0.331** |
| (Nie et al., 2023) | NMSE | 1.074 | 1.065 | 0.982 | 1.037 | 1.541 | 1.580 | 1.318 | 1.380 | 1.245 | 1.093 | 1.130 | 1.079 | 1.549 | 1.471 | 1.485 | 1.612 | 0.320 | 0.267 | 0.292 | 0.339 | 2.777 | 2.159 | 1.987 | 1.818 |
| | R | 0.849 | 0.900 | 0.916 | 0.900 | 0.918 | 0.901 | 0.907 | 0.866 | 0.867 | 0.803 | 0.789 | 0.861 | 0.834 | 0.845 | 0.892 | 0.917 | 0.777 | 0.761 | 0.739 | 0.900 | 0.931 | 0.873 | 0.850 | 0.900 |
| TimesNet | MSE | 0.389 | 0.460 | 0.487 | 0.641 | 0.337 | 0.405 | 0.399 | 0.447 | 0.334 | 0.414 | 0.429 | 0.482 | 0.189 | 0.239 | 0.320 | 0.383 | 0.168 | 0.189 | 0.209 | 0.305 | **0.169** | 0.220 | 0.272 | 0.334 |
| | MAE | 0.412 | 0.456 | 0.477 | 0.582 | 0.371 | 0.424 | 0.433 | 0.463 | 0.375 | 0.414 | 0.434 | 0.477 | 0.266 | 0.306 | 0.357 | 0.408 | 0.272 | 0.291 | 0.308 | 0.382 | 0.219 | 0.265 | 0.301 | 0.350 |
| (Wu et al., 2023) | NMSE | 1.205 | 1.227 | 1.113 | 1.341 | 1.952 | 2.115 | 1.556 | 1.527 | 1.391 | 1.424 | 1.352 | 1.361 | 1.714 | 1.621 | 1.933 | 2.012 | 0.316 | 0.345 | 0.350 | 0.482 | 2.555 | 2.531 | 2.294 | 2.035 |
| | R | 0.869 | 0.881 | 0.915 | 0.876 | 0.886 | 0.920 | 0.911 | 0.864 | 0.813 | 0.652 | 0.741 | 0.835 | 0.904 | 0.908 | 0.863 | 0.909 | 0.735 | 0.809 | 0.737 | 0.969 | 0.912 | 0.903 | 0.868 | 0.865 |
| Average | MSE | 0.386 | 0.438 | 0.465 | 0.589 | 0.310 | 0.379 | 0.408 | 0.491 | 0.333 | 0.353 | 0.385 | 0.449 | 0.183 | 0.239 | 0.289 | 0.374 | 0.166 | 0.163 | 0.177 | 0.224 | 0.177 | 0.212 | 0.257 | 0.332 |
| | MAE | 0.403 | 0.435 | 0.446 | 0.550 | 0.357 | 0.406 | 0.446 | 0.485 | 0.369 | 0.379 | 0.402 | 0.446 | 0.267 | 0.306 | 0.339 | 0.399 | 0.257 | 0.257 | 0.273 | 0.316 | 0.224 | 0.257 | 0.295 | 0.349 |
| | NMSE | 1.149 | 1.120 | 1.105 | 1.321 | 1.912 | 1.863 | 1.738 | 1.936 | 1.321 | 1.199 | 1.193 | 1.184 | 1.608 | 1.599 | 1.626 | 1.805 | 0.415 | 0.366 | 0.356 | 0.432 | 2.814 | 2.421 | 2.179 | 2.047 |
| Predictability | $MSE_{lb}$ | 0.354 | 0.417 | 0.404 | 0.412 | 0.298 | 0.360 | 0.309 | 0.356 | 0.228 | 0.307 | 0.513 | 0.436 | 0.175 | 0.248 | 0.295 | 0.361 | 0.239 | 0.219 | 0.167 | 0.241 | 0.185 | 0.244 | 0.278 | 0.317 |
| | $\mathcal{P}$ | 0.422 | 0.379 | 0.368 | 0.389 | 0.305 | 0.270 | 0.302 | 0.267 | 0.590 | 0.460 | 0.268 | 0.356 | 0.415 | 0.315 | 0.230 | 0.271 | 0.751 | 0.755 | 0.829 | 0.777 | 0.345 | 0.240 | 0.228 | 0.289 |

used datasets. To ensure a fair comparison, we follow a strictly controlled protocol: (i) the forecast horizon and lookback window are fixed and identical across all models; and (ii) the common "drop–last" heuristic is disabled to avoid subtle sampling biases. For correlation analysis, we compute the Pearson coefficient $R$ between each model's empirical MSE and the estimated linear lower bound $MSE_{lb}$ over the test set, aggregating across samples and variables to obtain a global summary statistic.

As shown in Table 1, $MSE_{lb}$ aligns closely with the realized errors of diverse forecasters across datasets and horizons, with Pearson correlations typically around $R \geq 0.8$. This supports the intended interpretation of $MSE_{lb}$: while it is formally a lower bound for linear time-invariant predictors, it functions empirically as a reliable *instance-difficulty reference*, indicating where forecasting is systematically easier or harder given the available history. In particular, instances deemed hard by our metric (high $MSE_{lb}$, low $\mathcal{P}$) consistently yield larger prediction errors across all tested architectures.

Beyond dataset-level averages, we observe substantial within-dataset heterogeneity: both predictability and empirical error vary markedly across variables, suggesting that a single aggregate score can obscure important structure. Figure 2 visualizes the relationship between $MSE_{lb}$ and iTransformer's realized MSE at the channel level for Weather and ECL. The near-linear trend indicates that the proposed estimate can localize difficulty at fine granularity, separating channels that are intrinsically hard to forecast from those that are structurally predictable.

To compare difficulty across datasets, we further aggregate

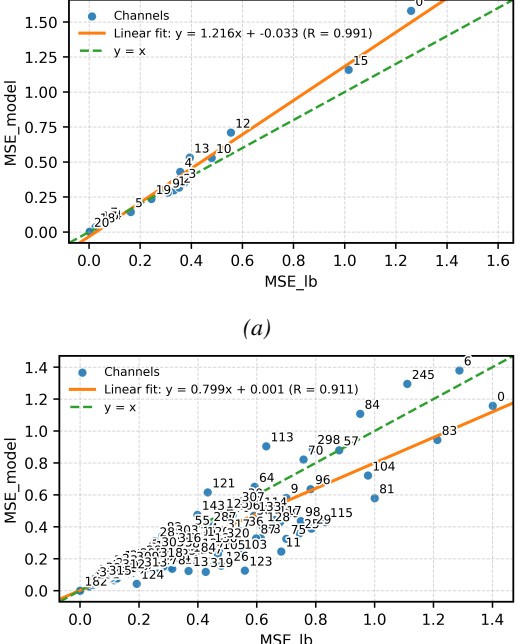

*Figure 2.* Per-variable scatter plots on Weather (a) and ECL (b) comparing the estimated MSE lower bound ($MSE_{lb}$) with iTransformer's prediction error $(MSE)_{model}$.

over horizons and report, for each dataset, the mean±std of SCP $\mathcal{P}$ together with the realized NMSE (Fig. 1b). The relationship is strongly inverse: datasets with higher $\mathcal{P}$ tend to exhibit lower NMSE across a broad range of architectures. For example, ECL consistently shows higher $\mathcal{P}$ and lower

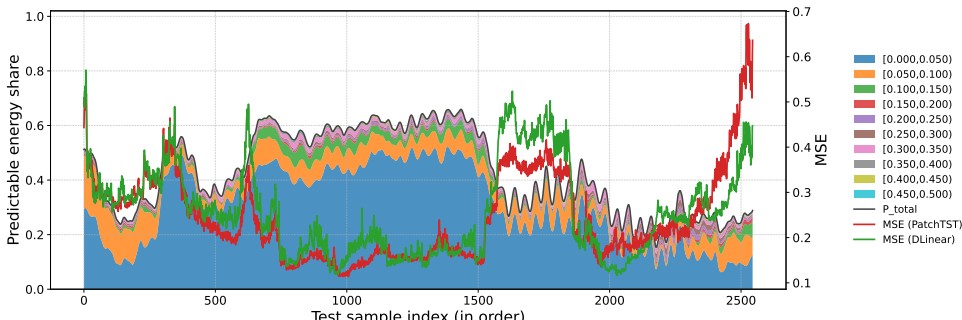

*Figure 3.* Visualizing Predictability Drift. ETTh1 test set, channel 1, horizon $N = 336$. Relationship between per-sample linearly predictable energy (decomposed by frequency band as a share of total) and the corresponding MSE of DLinear and PatchTST.

NMSE, whereas Weather shows lower $\mathcal{P}$ and higher NMSE, matching the difficulty ranking induced by the proposed proxy.

Ultimately, the observation that no single architecture dominates across all settings underscores a critical reality: performance is inextricably linked to the variability of the exploitable information across time, variables, and spectral components. These findings necessitate the shift from simple leaderboards to the difficulty-aware diagnostic analyses presented in the following sections.

### 5.3. Time-Varying Predictability

Standard evaluation metrics average errors over an entire test set, implicitly treating the forecasting task as statistically stationary. For real-world time series, this assumption is often violated: the amount of exploitable information can change over time, even within a single variable.

Figure 3 reveals a clear coupling between model error and the available predictable energy. When the total predictable share drops, or when the dominant predictable bands shift across time, the forecasting error rises sharply for both models. This shows that predictability is not a fixed property of a dataset; instead, the task alternates over time between easier and harder regimes. We refer to this time variation as *predictability drift*.

These observations also clarify why aggregate test-set statistics can be misleading: performance differences are partially confounded by the changing difficulty of the evaluated instances. They motivate difficulty-aware evaluation protocols, which can separate model limitations from fluctuations in the data and provide more actionable guidance for model development.

### 5.4. Band-wise Evaluation

To obtain a more fine-grained view of model behavior, we use the Linear Utilization Ratio (LUR) to analyze forecasting performance in the frequency domain. Specifically, we

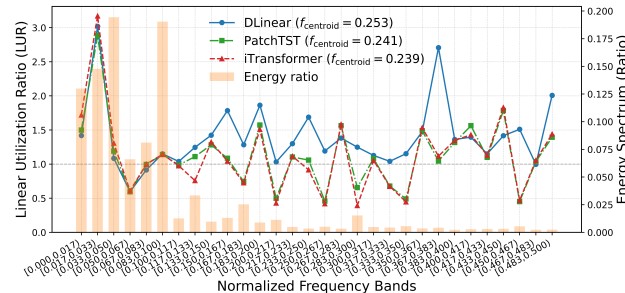

*Figure 4.* Band-wise analysis on ETTh1, representative channel. Normalized energy shares and LUR across frequency bands for three models.

partition the spectrum into disjoint frequency bands and report (i) the band's share of total target energy and (ii) the corresponding LUR for several representative models.

Figure 4 highlights clear architectural differences. In the low-frequency bands, which contain most of the signal energy and typically host the most stable structure, all three models exhibit high utilization. Their LUR values broadly follow the energy distribution. This suggests that each model is able to capture a substantial portion of the dominant components. Within these bands, PatchTST and iTransformer achieve slightly higher LUR than DLinear, indicating more effective extraction of linearly predictable component from the same history.

The contrast becomes more pronounced in the higher-frequency bands. Here, DLinear attains noticeably larger LUR than PatchTST and iTransformer. One plausible explanation is that the linear baseline tends to allocate capacity broadly across the spectrum, whereas Transformer-style models behave more selectively, prioritizing low-frequency components that are both higher-energy and typically more predictable, while de-emphasizing bands that are lower-energy and often dominated by irregular fluctuations. This demonstrates their more sophisticated inductive bias for typical time-series data.

## 5.5. Predictability-aware Evaluation

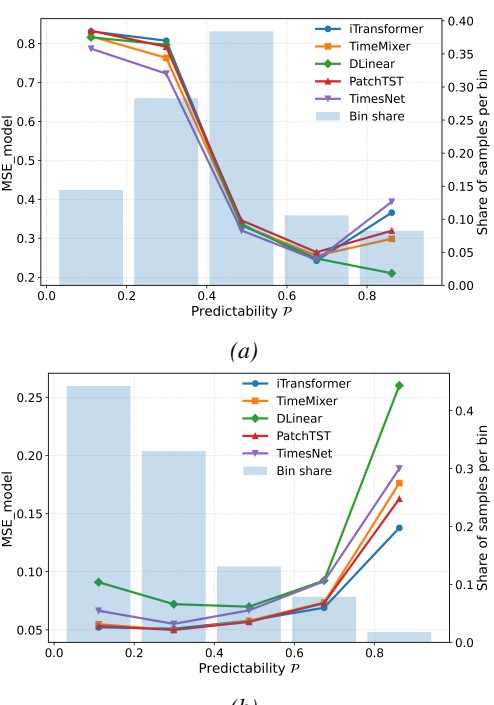

*(a)*

*(b)*

*Figure 5.* ETTh1 dataset with forecasting length $N = 96$: per-channel evaluation stratified by predictability $\mathcal{P}$. Samples are grouped into equal-width $\mathcal{P}$ bins; each point reports the mean MSE within the bin.

Although most models attain a similar average MSE (around 0.38) on ETTh1 at horizon $N{=}96$, this aggregate score can mask meaningful differences in where models succeed or fail. To expose these behaviors, we stratify the test set by the instance-level predictability score $\mathcal{P}$ and evaluate performance within equal-width $\mathcal{P}$ bins (Fig. 5).

The stratified results reveal clear architectural trade-offs. In Fig. 5a, nonlinear forecasters (e.g., TimesNet) achieve lower error in the low-$\mathcal{P}$ regime, which corresponds to hard samples with limited linearly exploitable structure, whereas DLinear becomes more competitive and can even dominate in the high-$\mathcal{P}$ regime, where samples are easier and linear information is abundant. In Fig. 5b, where the distribution is skewed toward low $\mathcal{P}$, the three high-capacity models (iTransformer, TimeMixer, PatchTST) show similar errors, while DLinear degrades more noticeably on the hardest bins, consistent with a capacity and expressivity limitation under low linear predictability. These contrasts highlight complementary strengths across architectures: nonlinear models excel when the linear predictable signal is scarce, whereas linear models are highly competitive when predictability is high.

*Table 2.* SCP and linear MSE lower bound (MSE$_{\text{lb}}$) under different Welch configurations.

| Parameter | Value | SCP (mean ± std) | MSE$_{\text{lb}}$ (mean ± std) |
|---|---|---|---|
| Window-length fraction ($L_w/N$) | 0.25 | $0.344 \pm 0.109$ | $0.186 \pm 0.133$ |
| | 0.30 | $0.345 \pm 0.110$ | $0.186 \pm 0.133$ |
| | 0.35 | $0.367 \pm 0.108$ | $0.183 \pm 0.133$ |
| Overlap ($\rho$) | 0.45 | $0.362 \pm 0.109$ | $0.184 \pm 0.134$ |
| | 0.50 | $0.344 \pm 0.109$ | $0.185 \pm 0.133$ |
| | 0.55 | $0.351 \pm 0.109$ | $0.185 \pm 0.135$ |
| Window type | Hann | $0.345 \pm 0.110$ | $0.186 \pm 0.134$ |
| | Hamming | $0.351 \pm 0.110$ | $0.185 \pm 0.133$ |
| | Blackman | $0.335 \pm 0.110$ | $0.187 \pm 0.135$ |

## 5.6. Sensitivity to Welch Parameters

We evaluate the sensitivity of SCP to the Welch hyperparameters. On the Weather dataset with horizon fixed to $N{=}96$, we vary three factors: the window-length fraction $L_w/N$, the overlap ratio $\rho$, and the tapering window (Hann, Hamming, Blackman). We use $(L_w/N, \rho, \text{window}) = (0.25, 0.5, \text{Hann})$ as the default setting, and vary one hyperparameter at a time while keeping the others fixed.

As reported in Table 2, both the mean SCP and the mean MSE$_{\text{lb}}$ change only modestly across the tested ranges, suggesting that the estimator is stable under standard spectral configurations. The comparatively large standard deviations primarily reflect genuine heterogeneity in the data, i.e., predictability varies substantially across instances.

## 5.7. Variable History Window

*Table 3.* Effect of history window length $N_x$ on model error (MSE, MSE$_{\text{lb}}$) and correlation $R$ on ETTh1 ($N_y = 336$).

| Model | Metric | History length $N_x$ | | | |
|---|---|---|---|---|---|
| | | 96 | 192 | 336 | 720 |
| iTransformer | MSE | 0.491 | 0.479 | 0.471 | 0.480 |
| | $R$ | 0.808 | 0.842 | 0.899 | 0.832 |
| DLinear | MSE | 0.491 | 0.480 | 0.447 | 0.449 |
| | $R$ | 0.788 | 0.840 | 0.872 | 0.853 |
| Predictibility | MSE$_{\text{lb}}$ | 0.431 | 0.433 | 0.404 | 0.424 |

We conduct this study on ETTh1 with a fixed prediction horizon $N_y{=}336$ and varying history lengths $N_x \in \{96, 192, 336, 720\}$, evaluating both iTransformer and DLinear. Table 3 summarizes the results.

As $N_x$ changes, the absolute MSE of both models varies, reflecting the practical sensitivity of forecasting accuracy to the amount of available context. In contrast, the Pearson correlation $R$ between MSE$_{\text{lb}}$ and per-sample errors remains consistently high across history lengths (typically $R \geq 0.80$). This suggests that SCP captures a stable notion of instance difficulty that is not tightly coupled to a specific choice of $N_x$, even when the models' absolute accuracy depends on the history window length.

# 6. Conclusion

Standard forecasting metrics conflate model limitations with instance difficulty, obscuring why errors occur. We proposed a predictability-aligned diagnostic framework based on spectral coherence. SCP provides an efficient ($\mathcal{O}(N \log N)$) instance-level difficulty reference and a corresponding linear MSE lower bound, while LUR offers a frequency-resolved measure of how effectively a model exploits linearly predictable structure. Experiments on synthetic and real benchmarks show that SCP/MSE$_{\text{lb}}$ are well-calibrated and strongly aligned with realized errors, enabling difficulty-aware evaluation. We further reveal pronounced time- and variable-level variation in predictability (predictability drift) and show that stratifying results by SCP exposes complementary architectural strengths. In summary, we advocate moving beyond model ranking toward predictability-aware diagnostics that enable fairer comparisons and more actionable understanding of model behavior.

# Impact Statement

This paper presents a diagnostic framework for time-series forecasting evaluation by quantifying instance-level predictability and analyzing how models utilize linearly predictable structure. The primary intended impact is to improve fairness, transparency, and interpretability in model comparison, and to reduce wasted computation by distinguishing intrinsically hard instances from model deficiencies.

Beyond evaluation, the proposed metrics provide a rigorous basis to guide future architectural innovations and training strategies. Specifically, our predictability scores can enable the design of adaptive architectures, such as Mixture-of-Experts (MoE) systems that dynamically route samples based on difficulty, as well as data-efficient training paradigms like predictability-aware curriculum learning and hard sample mining. We do not anticipate broader societal risks beyond those commonly associated with general time-series forecasting applications.

# Acknowledgements

This work was supported in part by the National Natural Science Foundation of China under Grants U24B20180, U23B2030, and 62476152.

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

# A. Experimental Setup

## A.1. Toy Study

We synthesize a multiband Gaussian process and evaluate linear forecasting under controlled band-limited noise. Signals are split into history/future with boundary-paired segments of length $N_p = 512$ from a total length $2N$ with $N = 1024$. Power spectra and coherences are estimated via Welch's method (Hann window) with $n_{\mathrm{perseg}} = 256$ and $n_{\mathrm{overlap}} = 128$. The forecaster is a causal FIR least-squares filter (Wiener approximation) of length $L_{\mathrm{FIR}} = 64$ with ridge $10^{-6}$. The base process has four spectral peaks at rFFT bins $\{32, 96, 192, 384\}$ with widths $\{6, 10, 14, 18\}$ and amplitudes $\{3.0, 2.0, 1.5, 1.0\}$. We sweep noise levels $\{0, 0.25, 0.5, 1.0, 2.0, 4.0\}$ on a single band (index 1 in the plot) and average over 3 trials, reporting model MSE, $\mathrm{MSE}_{\mathrm{lb}}$, and SCP.

## A.2. Backbone

We evaluate five state-of-the-art backbones spanning diverse architectures: Transformer-based (iTransformer (Liu et al., 2024), PatchTST (Nie et al., 2023)), MLP-based (DLinear (Zeng et al., 2023), TimeMixer (Wang et al., 2024)), and CNN-based (TimesNet (Wu et al., 2023)). We adopt the official implementations and recommended hyperparameters from their repositories. To ensure strict comparability, we fix the forecasting horizon and enforce equal input and output lengths for all backbones (no "drop-last"), using identical preprocessing and dataset splits across models.

## A.3. Datasets

We conduct experiments on eight standard long-horizon multivariate forecasting benchmarks: ETTh1, ETTh2, ETTm1, ETTm2, ECL, Weather, Traffic, and ILI. These datasets cover electricity systems, meteorology, transportation, and epidemiology, and are widely used in recent long-horizon time series forecasting studies. Table 4 summarizes the basic statistics and forecasting horizon settings used in this extended evaluation.

*Table 4.* Detailed descriptions of the datasets used in our extended evaluation. "Number of variables" gives the dimensionality of each dataset. "Dataset size" denotes the total number of time points in the training, validation, and test splits. "Prediction length" denotes the forecasting horizon; four horizon settings are used for each dataset. "Frequency" is the sampling interval.

| Dataset | Dim | Prediction Length | Dataset Size | Frequency | Information |
|---|---|---|---|---|---|
| ETTh1, ETTh2 | 7 | $\{96, 192, 336, 720\}$ | (8545, 2881, 2881) | Hourly | Electricity |
| ETTm1, ETTm2 | 7 | $\{96, 192, 336, 720\}$ | (34465, 11521, 11521) | 15min | Electricity |
| ECL | 321 | $\{96, 192, 336, 720\}$ | (18317, 2633, 5261) | Hourly | Electricity |
| Weather | 21 | $\{96, 192, 336, 720\}$ | (36792, 5271, 10540) | 10min | Weather |
| Traffic | 862 | $\{96, 192, 336, 720\}$ | (12185, 1757, 3509) | Hourly | Transportation |
| ILI | 7 | $\{60, 72\}$ | (617, 74, 170) | Weekly | Epidemiology |

## A.4. Time-to-Frequency

For each test instance we split the sequence into history $x$ and future $y$ (equal lengths by default), remove sample means, and estimate power and cross-spectra with Welch's method using identical settings for $x$, $y$, and (when available) $\hat{y}$: Hann window with length $n_{\mathrm{win}} = \lfloor 0.25N \rfloor$, 50% overlap, and real FFT on the one-sided grid $\mathcal{F}$ with variance-preserving normalization. We form squared coherences with a small ridge $\varepsilon$ for stability, compute the residual spectrum to obtain the linear lower bound $\mathrm{MSE}_{\mathrm{lb}}$ and predictability $\mathcal{P} = 1 - \mathrm{MSE}_{\mathrm{lb}}/\mathrm{Var}(y)$, and derive utilization metrics (global or band-wise) via target-power–weighted aggregation of $\gamma^2_{y\hat{y}}$ and $\gamma^2_{yx}$.

# B. Method Extensions

In the main text, we focus on univariate predictability and its linear component, and use spectral coherence to quantify the linearly exploitable information between a history segment $x$ and its future $y$. This choice is deliberate: the univariate linear formulation yields a conservative and highly interpretable difficulty reference, requires minimal modeling assumptions, and can be implemented efficiently with standard spectral estimators. As a result, it provides a practical diagnostic baseline that is easy to reproduce and robust in the finite-sample, non-stationary regimes common in real-world forecasting benchmarks.

At the same time, the proposed framework is not limited to this setting. It naturally supports extensions to multivariate time

---

**Algorithm 3** Multivariate Spectral Coherence Predictability ($\text{SCP}_{\text{multi}}$)

---

**Require:** History $\boldsymbol{x} \in \mathbb{R}^{d_x \times N}$, future $\boldsymbol{y} \in \mathbb{R}^{d_y \times N}$; Welch parameters (window, length, overlap); stability constant $\varepsilon > 0$; optional frequency band $\mathcal{F}_b$.
**Ensure:** Multivariate MSE lower bound $\text{MSE}_{\text{lb}}^{\text{multi}}$ and predictability $\mathcal{P}_{xy}^{\text{multi}}$.
1: **Mean removal:** $\Delta^2 \leftarrow \|\boldsymbol{\mu}_y - \boldsymbol{\mu}_x\|_2^2$; $\boldsymbol{x} \leftarrow \boldsymbol{x} - \boldsymbol{\mu}_x$, $\boldsymbol{y} \leftarrow \boldsymbol{y} - \boldsymbol{\mu}_y$.
2: **Welch spectra:** Compute matrix-valued PSDs $\widehat{S}_{xx}(f)$, $\widehat{S}_{yy}(f)$ and CPSD $\widehat{S}_{xy}(f)$ on $\mathcal{F}$; set $\widehat{S}_{yx}(f) = \widehat{S}_{xy}(f)^H$.
3: **Multichannel Wiener spectra:**

$$\widehat{S}_{\hat{y}\hat{y}}(f) = \widehat{S}_{yx}(f)\big(\widehat{S}_{xx}(f) + \varepsilon I_{d_x}\big)^{-1}\widehat{S}_{xy}(f), \quad \widehat{S}_e(f) = \widehat{S}_{yy}(f) - \widehat{S}_{\hat{y}\hat{y}}(f).$$

4: **Frequency set:** $\mathcal{F}_\star \leftarrow \mathcal{F}_b$ if $\mathcal{F}_b$ is provided; otherwise $\mathcal{F}_\star \leftarrow \mathcal{F}$.
5: **Aggregate:**
$$\widehat{\text{Var}}(\boldsymbol{y}) \;\leftarrow\; \sum_{f \in \mathcal{F}_\star} \text{tr}\,\widehat{S}_{yy}(f), \qquad \text{MSE}_{\text{lb}}^{\text{multi}} \;\leftarrow\; \Delta^2 + \sum_{f \in \mathcal{F}_\star} \text{tr}\,\widehat{S}_e(f).$$

6: **Predictability:** $\mathcal{P}_{xy}^{\text{multi}} \leftarrow 1 - \text{MSE}_{\text{lb}}^{\text{multi}}/\widehat{\text{Var}}(\boldsymbol{y})$.
7: **Return:** $\text{MSE}_{\text{lb}}^{\text{multi}}$, $\mathcal{P}_{xy}^{\text{multi}}$.

---

series, where cross-channel dependencies can be incorporated through matrix-valued spectral estimates and coherence-based diagnostics, as well as nonlinear variants that aim to capture dependence beyond linear time-invariant structure. These extensions can be beneficial when cross-variable interactions or nonlinear dynamics carry substantial predictive signal.

We defer the detailed derivations and algorithmic variants to the appendix to keep the main presentation general and easy to adopt. The multivariate and nonlinear versions introduce additional estimation choices (e.g., conditioning strategies, regularization, or nonlinear dependence measures) that are not required for our core claims and empirical findings, but are important for completeness and for practitioners who wish to apply the framework in richer settings. Below, we summarize these extensions and provide the corresponding formulations.

### B.1. Multivariate Extension

B.1.1. MULTIVARIATE SCP

We extend the univariate SCP in Sec. 4.1 to multivariate histories and futures with input dimensionality $d_x$ and output dimensionality $d_y$. Let $\boldsymbol{x}_t \in \mathbb{R}^{d_x}$ and $\boldsymbol{y}_t \in \mathbb{R}^{d_y}$ denote a length-$N$ history–future pair, and let $\boldsymbol{x} = (\boldsymbol{x}_1, \ldots, \boldsymbol{x}_N) \in \mathbb{R}^{d_x \times N}$, $\boldsymbol{y} = (\boldsymbol{y}_1, \ldots, \boldsymbol{y}_N) \in \mathbb{R}^{d_y \times N}$. Using Welch's method with shared parameters for all components, we compute multivariate power spectral density (PSD) and cross–power spectral density (CPSD) matrices on a discrete frequency grid $\mathcal{F}$:

$$\widehat{S}_{xx}(f) \in \mathbb{C}^{d_x \times d_x}, \qquad\qquad \widehat{S}_{yy}(f) \in \mathbb{C}^{d_y \times d_y}, \qquad\qquad \widehat{S}_{xy}(f) \in \mathbb{C}^{d_x \times d_y}, \qquad (13)$$

and set $\widehat{S}_{yx}(f) = \widehat{S}_{xy}(f)^H$, where $H$ denotes the Hermitian transpose.

At frequency $f$, the optimal linear time–invariant predictor from $\boldsymbol{x}$ to $\boldsymbol{y}$ in the least-squares sense has transfer matrix

$$H(f) \;=\; \widehat{S}_{yx}(f)\Big(\widehat{S}_{xx}(f) + \varepsilon I_{d_x}\Big)^{-1}, \qquad (14)$$

where $\varepsilon > 0$ is the same Tikhonov regularization as in Eq. (4), and $I_{d_x}$ is the $d_x \times d_x$ identity matrix. The spectrum of the linearly predictable component of $\boldsymbol{y}$ is then

$$\widehat{S}_{\hat{y}\hat{y}}(f) \;=\; H(f)\,\widehat{S}_{xx}(f)\,H(f)^H \;=\; \widehat{S}_{yx}(f)\Big(\widehat{S}_{xx}(f) + \varepsilon I_{d_x}\Big)^{-1}\widehat{S}_{xy}(f) \;\in\; \mathbb{C}^{d_y \times d_y}. \qquad (15)$$

In the scalar case $d_x = d_y = 1$, Eq. (15) reduces to $\widehat{S}_{\hat{y}\hat{y}}(f) = |\widehat{S}_{xy}(f)|^2/\big(\widehat{S}_{xx}(f) + \varepsilon\big)$, which coincides with the univariate expression $\gamma_{xy}^2(f)\,\widehat{S}_{yy}(f)$ in Eq. (4).

The residual spectrum matrix is
$$\widehat{S}_e(f) \;=\; \widehat{S}_{yy}(f) - \widehat{S}_{\hat{y}\hat{y}}(f), \qquad \forall f \in \mathcal{F}. \qquad (16)$$

---

**Algorithm 4** Multivariate Linear Utilization Ratio (LUR$_{\text{multi}}$)

---

**Require:** History $\boldsymbol{x} \in \mathbb{R}^{d_x \times N}$, future $\boldsymbol{y} \in \mathbb{R}^{d_y \times N}$, prediction $\hat{\boldsymbol{y}} \in \mathbb{R}^{d_y \times N}$; Welch parameters (window, length, overlap); stability $\varepsilon > 0$; optional band $\mathcal{F}_b$.

**Ensure:** Multivariate model–explained power $P_{\text{model}}$, linear–explainable power $P_{\text{linear}}$, and utilization ratio $\text{LUR}^{\text{multi}}$.

1: **Mean removal:** $\boldsymbol{x} \leftarrow \boldsymbol{x} - \text{mean}(\boldsymbol{x})$; $\boldsymbol{y} \leftarrow \boldsymbol{y} - \text{mean}(\boldsymbol{y})$; $\hat{\boldsymbol{y}} \leftarrow \hat{\boldsymbol{y}} - \text{mean}(\hat{\boldsymbol{y}})$.

2: **Welch spectra:** Compute $\widehat{S}_{xx}(f)$, $\widehat{S}_{yy}(f)$, $\widehat{S}_{\hat{y}\hat{y}}^{\text{pred}}(f)$, $\widehat{S}_{xy}(f)$, $\widehat{S}_{y\hat{y}}(f)$ on $\mathcal{F}$; set $\widehat{S}_{yx}(f) = \widehat{S}_{xy}(f)^H$ and $\widehat{S}_{\hat{y}y}(f) = \widehat{S}_{y\hat{y}}(f)^H$.

3: **Linear limit (per frequency):**

$$\widehat{S}_{\hat{y}\hat{y}}(f) \leftarrow \widehat{S}_{yx}(f)\big(\widehat{S}_{xx}(f) + \varepsilon I_{d_x}\big)^{-1}\widehat{S}_{xy}(f), \quad P_{\text{linear}}(f) \leftarrow \text{tr}\,\widehat{S}_{\hat{y}\hat{y}}(f).$$

4: **Model–explained power (per frequency):**

$$P_{\text{model}}(f) \leftarrow \text{tr}\Big(\widehat{S}_{y\hat{y}}(f)\big(\widehat{S}_{\hat{y}\hat{y}}^{\text{pred}}(f) + \varepsilon I_{d_y}\big)^{-1}\widehat{S}_{\hat{y}y}(f)\Big).$$

5: **Frequency set:** $\mathcal{F}_\star \leftarrow \mathcal{F}_b$ if a band $\mathcal{F}_b$ is provided; otherwise $\mathcal{F}_\star \leftarrow \mathcal{F}$.

6: **Aggregation:**

$$P_{\text{linear}} \leftarrow \sum_{f \in \mathcal{F}_\star} P_{\text{linear}}(f), \qquad P_{\text{model}} \leftarrow \sum_{f \in \mathcal{F}_\star} P_{\text{model}}(f).$$

7: **LUR ratio:** $\text{LUR}^{\text{multi}} \leftarrow P_{\text{model}}/P_{\text{linear}}$.

8: **Return:** $P_{\text{model}}$, $P_{\text{linear}}$, $\text{LUR}^{\text{multi}}$.

---

Since $\widehat{S}_{\hat{y}\hat{y}}(f)$ is the least-squares projection of $\widehat{S}_{yy}(f)$ onto the subspace linearly spanned by $\boldsymbol{x}$, the true residual spectrum is positive semidefinite, and the regularization $\varepsilon I_{d_x}$ stabilizes this property numerically. Let the estimated total variance (total power) of $\boldsymbol{y}$ be the trace–aggregated spectrum

$$\widehat{\text{Var}}(\boldsymbol{y}) \;=\; \sum_{f \in \mathcal{F}} \text{tr}\,\widehat{S}_{yy}(f), \tag{17}$$

where $\text{tr}(\cdot)$ denotes the matrix trace. Using the same frequency grid, the multivariate MSE lower bound induced by linear time–invariant predictors is

$$\text{MSE}_{\text{lb}}^{\text{multi}} \;=\; \Delta^2 \;+\; \sum_{f \in \mathcal{F}} \text{tr}\,\widehat{S}_e(f), \tag{18}$$

where $\Delta^2$ is the same boundary mean–shift term as in the univariate case, generalized to the $(d_x, d_y)$-dimensional setting.

The multivariate SCP is defined by normalizing the residual energy as in Eq. (8):

$$\mathcal{P}_{xy}^{\text{multi}} \;=\; 1 - \frac{\text{MSE}_{\text{lb}}^{\text{multi}}}{\widehat{\text{Var}}(\boldsymbol{y})} \;\in\; [0, 1]. \tag{19}$$

When $d_x = d_y = 1$, Eq. (19) reduces exactly to the univariate SCP in Eq. (8).

### B.1.2. MULTIVARIATE LUR

The spectrum of the linearly predictable component in Eq. (15) induces the linear–explainable power

$$P_{\text{linear}}(f) \;=\; \text{tr}\,\widehat{S}_{\hat{y}\hat{y}}(f) \;=\; \text{tr}\Big(\widehat{S}_{yx}(f)\big(\widehat{S}_{xx}(f) + \varepsilon I_{d_x}\big)^{-1}\widehat{S}_{xy}(f)\Big). \tag{20}$$

For the model, we form the auto- and cross-spectra of the prediction,

$$\widehat{S}_{\hat{y}\hat{y}}^{\text{pred}}(f) \in \mathbb{C}^{d_y \times d_y}, \qquad \widehat{S}_{y\hat{y}}(f) \in \mathbb{C}^{d_y \times d_y}, \qquad \widehat{S}_{\hat{y}y}(f) = \widehat{S}_{y\hat{y}}(f)^H, \tag{21}$$

and define the model–explained power via the optimal linear projection of $\boldsymbol{y}$ onto the subspace spanned by $\hat{\boldsymbol{y}}$:

$$P_{\text{model}}(f) \;=\; \text{tr}\Big(\widehat{S}_{y\hat{y}}(f)\big(\widehat{S}_{\hat{y}\hat{y}}^{\text{pred}}(f) + \varepsilon I_{d_y}\big)^{-1}\widehat{S}_{\hat{y}y}(f)\Big). \tag{22}$$

Aggregating over the discrete frequency domain $\mathcal{F}$,

$$P_{\text{linear}} = \sum_{f \in \mathcal{F}} P_{\text{linear}}(f), \qquad P_{\text{model}} = \sum_{f \in \mathcal{F}} P_{\text{model}}(f), \tag{23}$$

and normalizing as in Sec. 4.2 gives the multivariate linear utilization ratio

$$\text{LUR}^{\text{multi}} = \frac{P_{\text{model}}}{P_{\text{linear}}}. \tag{24}$$

When $d_x = d_y = 1$, these expressions reduce to the univariate definitions of $P_{\text{linear}}$, $P_{\text{model}}$, and LUR.

## B.2. Nonlinear Extension

The SCP framework is linear by construction: it characterizes the best linear time–invariant (LTI) predictor in the original observation space. To relax this restriction while preserving the same spectral machinery, we introduce a nonlinear feature map

$$\phi : \mathbb{R}^{d_x} \to \mathbb{R}^{d_z}, \qquad \boldsymbol{z}_t = \phi(\boldsymbol{x}_t) \in \mathbb{R}^{d_z}, \tag{25}$$

and apply multivariate SCP in the resulting feature space. We then form the feature sequence $\boldsymbol{z} = (\boldsymbol{z}_1, \ldots, \boldsymbol{z}_N) \in \mathbb{R}^{d_z \times N}$. The map $\phi$ can use explicit nonlinear features (e.g., polynomial expansions or a shallow encoder), or be defined implicitly by a kernel $k(\boldsymbol{x}, \boldsymbol{x}') = \langle \phi(\boldsymbol{x}), \phi(\boldsymbol{x}') \rangle$ in an RKHS. Using the same Welch configuration as before, we estimate the multivariate spectra

$$\widehat{S}_{zz}(f) \in \mathbb{C}^{d_z \times d_z}, \qquad \widehat{S}_{yy}(f) \in \mathbb{C}^{d_y \times d_y}, \qquad \widehat{S}_{yz}(f) \in \mathbb{C}^{d_y \times d_z}, \tag{26}$$

and set $\widehat{S}_{zy}(f) = \widehat{S}_{yz}(f)^H$.

In feature space, the optimal LTI predictor of $\boldsymbol{y}$ from $\boldsymbol{z}$ takes the same form as the multivariate Wiener filter in Eq. (14), but with $(\boldsymbol{x}, \widehat{S}_{xx})$ replaced by $(\boldsymbol{z}, \widehat{S}_{zz})$:

$$H_\phi(f) = \widehat{S}_{yz}(f) \Big( \widehat{S}_{zz}(f) + \varepsilon I_{d_z} \Big)^{-1}, \tag{27}$$

where $\varepsilon > 0$ is the same Tikhonov regularization as before and $I_{d_z}$ is the $d_z \times d_z$ identity. The spectrum of the component of $\boldsymbol{y}$ that is linearly predictable from the nonlinear features is

$$\widehat{S}_{\hat{y}\hat{y}}^{\text{ker}}(f) = H_\phi(f) \widehat{S}_{zz}(f) H_\phi(f)^H = \widehat{S}_{yz}(f) \Big( \widehat{S}_{zz}(f) + \varepsilon I_{d_z} \Big)^{-1} \widehat{S}_{zy}(f) \in \mathbb{C}^{d_y \times d_y}. \tag{28}$$

When $\phi$ is the identity map ($d_z = d_x$ and $\boldsymbol{z}_t = \boldsymbol{x}_t$), Eq. (28) reduces to the multivariate linear spectrum in Eq. (15).

The residual spectrum under the feature-space predictor is

$$\widehat{S}_e^{\text{ker}}(f) = \widehat{S}_{yy}(f) - \widehat{S}_{\hat{y}\hat{y}}^{\text{ker}}(f), \qquad \forall f \in \mathcal{F}, \tag{29}$$

which is positive semidefinite in the ideal (population) setting. Aggregating as in Eq. (17), the total variance of $\boldsymbol{y}$ and the corresponding nonlinear MSE lower bound are

$$\widehat{\text{Var}}(\boldsymbol{y}) = \sum_{f \in \mathcal{F}} \text{tr}\, \widehat{S}_{yy}(f), \qquad \text{MSE}_{\text{lb}}^{\text{ker}} = \Delta^2 + \sum_{f \in \mathcal{F}} \text{tr}\, \widehat{S}_e^{\text{ker}}(f), \tag{30}$$

where $\Delta^2$ is the same boundary mean–shift term used in Eq. (18), applied to the multivariate setting.

The nonlinear SCP is then obtained by normalizing the feature-space residual:

$$\mathcal{P}_{xy}^{\text{nonlin}} = 1 - \frac{\text{MSE}_{\text{lb}}^{\text{ker}}}{\widehat{\text{Var}}(\boldsymbol{y})}. \tag{31}$$

This quantity measures the fraction of future variance that is explainable by LTI predictors acting on the chosen nonlinear feature representation, providing a feature-dependent notion of nonlinear predictability.

**B.3. Variable History Window ($N_x \neq N_y$)**

Let $N_x$ and $N_y$ denote the history and future lengths used for SCP. The construction only requires that a contiguous history–future pair exists around the boundary; $N_x$ and $N_y$ need not coincide. Given a Welch segment length $L_{\mathrm{w}}$ and overlap ratio $\mathrm{overlap} \in [0, 1)$, the effective shift between consecutive segments is

$$\Delta = L_{\mathrm{w}}(1 - \mathrm{overlap}),$$

and the approximate number of Welch segments for a sequence of length $N$ is

$$K(N; L_{\mathrm{w}}, \mathrm{overlap}) \approx \left\lfloor \frac{N - L_{\mathrm{w}}}{L_{\mathrm{w}}(1 - \mathrm{overlap})} \right\rfloor + 1. \tag{32}$$

As a concrete example, consider a history window $N_x = 192$, a longer future horizon $N_y = 336$, and a Welch window $L_{\mathrm{w}} = 64$. With an overlap of $\mathrm{overlap} = 0.5$, the hop size is $\Delta = 64(1 - 0.5) = 32$, and the corresponding numbers of Welch segments are

$$K_x \approx K(192; 64, 0.5) = \left\lfloor \tfrac{192-64}{32} \right\rfloor + 1 = 5, \qquad K_y \approx K(336; 64, 0.5) = \left\lfloor \tfrac{336-64}{32} \right\rfloor + 1 = 9.$$

For each segment we form windowed signals $\boldsymbol{x}_k(t)$ and $\boldsymbol{y}_k(t)$ of length $L_{\mathrm{w}}$, compute their discrete Fourier transforms $X_k(f)$ and $Y_k(f)$, and define the auto-spectra by Welch averaging

$$\widehat{S}_{xx}(f) = \frac{1}{K_x} \sum_{k=1}^{K_x} |X_k(f)|^2, \qquad \widehat{S}_{yy}(f) = \frac{1}{K_y} \sum_{k=1}^{K_y} |Y_k(f)|^2.$$

The cross-spectrum is computed on the aligned history–future portion at the boundary: we use the last $K_{\mathrm{pair}} = \min(K_x, K_y)$ segments from the history and the first $K_{\mathrm{pair}}$ segments from the future, denote their transforms by $X_k^{(\mathrm{hist})}(f)$ and $Y_k^{(\mathrm{fut})}(f)$, and set

$$\widehat{S}_{xy}(f) = \frac{1}{K_{\mathrm{pair}}} \sum_{k=1}^{K_{\mathrm{pair}}} X_k^{(\mathrm{hist})}(f) \, \overline{Y_k^{(\mathrm{fut})}(f)}.$$

Thus the shorter side effectively limits $K_{\mathrm{pair}}$ and hence the stability of $\widehat{S}_{xy}(f)$, while additional segments on the longer side primarily reduce the variance of the marginal auto-spectra.

**B.4. Beyond Evaluation**

The predictability scores from SCP (and its multivariate / nonlinear variants) $\mathcal{P}_{xy} \in [0, 1]$ can be used not only for post-hoc analysis, but also to shape how data are selected and organized during training.

**Hard-example mining** For a dataset $\{(\boldsymbol{x}^{(i)}, \boldsymbol{y}^{(i)})\}_{i=1}^{M}$ with per-sample predictability $\mathcal{P}^{(i)} \equiv \mathcal{P}_{xy}(\boldsymbol{x}^{(i)}, \boldsymbol{y}^{(i)})$, we can directly use $\mathcal{P}^{(i)}$ to reweight the loss:

$$L = \sum_{i=1}^{M} w^{(i)} \ell(\hat{\boldsymbol{y}}^{(i)}, \boldsymbol{y}^{(i)}), \qquad w^{(i)} \propto (\mathcal{P}^{(i)})^\alpha, \tag{33}$$

with $\alpha < 0$. This up-weights intrinsically predictable segments and down-weights near-unpredictable ones that mainly contain irreducible noise.

**Curriculum learning** The same scores induce a simple curriculum over data difficulty. Let $\{\tau_s\}_{s=1}^{S}$ be a decreasing sequence of thresholds, $\tau_1 > \tau_2 > \cdots > \tau_S$. At stage $s$, we restrict training to

$$\mathcal{D}_s = \{ i : \mathcal{P}^{(i)} \geq \tau_s \}, \tag{34}$$

i.e., the model first sees highly predictable segments, and progressively incorporates samples with lower $\mathcal{P}^{(i)}$ as $s$ increases.

**Anomaly detection and change points**    On a time series stream, we compute predictability over a sliding window ending at time $t$, for example $\mathcal{P}_t = \mathcal{P}_{xy}(\boldsymbol{x}_{t-N+1:t}, \boldsymbol{y}_{t+1:t+N})$. Let $\mu_{\mathcal{P}}, \sigma_{\mathcal{P}}$ be the mean and standard deviation of $\mathcal{P}_t$ on a reference (normal) period. We flag $t$ as anomalous when

$$\left| \mathcal{P}_t - \mu_{\mathcal{P}} \right| > \kappa \, \sigma_{\mathcal{P}}, \tag{35}$$

with a chosen threshold $\kappa > 0$. Sudden drops or spikes in $\mathcal{P}_t$ indicate changes in intrinsic predictability, and thus potential regime shifts or anomalous behavior.

### B.5. Comparison with time-domain correlation diagnostics

Classical time-domain tools such as the autocorrelation function (ACF) provide a convenient way to visualize second-order structure by plotting correlation as a function of lag. ACF is particularly useful for qualitatively assessing periodicity and dependence decay. However, it is primarily a descriptive tool for the self-correlation of a single series. In particular, ACF does not directly quantify how well a future window can be linearly predicted from a past window under the MSE objective, especially in the multi-horizon, multivariate setting we consider.

In contrast, our SCP/LUR framework is explicitly constructed around the past–future prediction task. SCP is derived from the cross-spectral density and coherence between the history and future segments, and measures the fraction of the future variance that is linearly explainable from the observed history, yielding an MSE-aligned notion of intrinsic predictability. LUR further decomposes this explainable energy across frequency bands, revealing which parts of the spectrum are well captured or systematically missed by a given model.

A simple example illustrates the difference between naive time-domain correlation and spectral coherence. Consider two noiseless signals

$$x_t = \sin(\omega_0 t), \qquad y_t = \cos(\omega_0 t).$$

Here $y_t$ is a phase-shifted version of $x_t$, obtained by a linear time-invariant transformation. In other words, $y$ is perfectly linearly predictable from $x$.

If we look only at the zero-lag Pearson correlation

$$\rho_{xy}(0) = \mathrm{corr}(x_t, y_t),$$

and average over many periods by treating $\theta = \omega_0 t$ as uniform on $[0, 2\pi]$, we obtain

$$\mathbb{E}[\sin\theta\cos\theta] = 0,$$

hence $\rho_{xy}(0) = 0$. A time-domain diagnostic based solely on zero-lag correlation would therefore suggest that $x$ and $y$ are "unrelated", even though $y$ is deterministically generated from $x$ by a linear filter.

In the frequency domain, both $x$ and $y$ have all their energy concentrated at the same frequency $\omega_0$. Their cross-spectrum at $\omega_0$ differs only by a constant phase factor, so the squared coherence

$$\gamma^2(\omega_0) = \frac{|S_{xy}(\omega_0)|^2}{S_{xx}(\omega_0)\, S_{yy}(\omega_0)}$$

evaluates to $\gamma^2(\omega_0) = 1$. In our framework, this implies a linear MSE lower bound of zero and SCP equal to one: the spectral diagnostic correctly recognizes that $y$ is fully predictable from $x$ despite the phase shift. This example highlights that simple time-domain summaries such as zero-lag correlation can miss strong linear predictability when phase shifts or distributed lags are present, whereas coherence (and thus SCP) aggregates information over all lags at each frequency and is invariant to such shifts.

## C. Supplementary Experiments

### C.1. Sensitivity to Frequency-band Partitioning

As illustrated in Figures 6a and 6b, changing the band boundaries affects the absolute LUR values within each band, but the qualitative conclusions remain unchanged. Across all configurations, iTransformer consistently achieves higher LUR in the low-frequency region where most signal energy concentrates, whereas DLinear performs better in the high-frequency bands. The frequency centroid $f_{\mathrm{centroid}}$ exhibits the same ordering: DLinear attains the largest centroid, while PatchTST and iTransformer remain close.

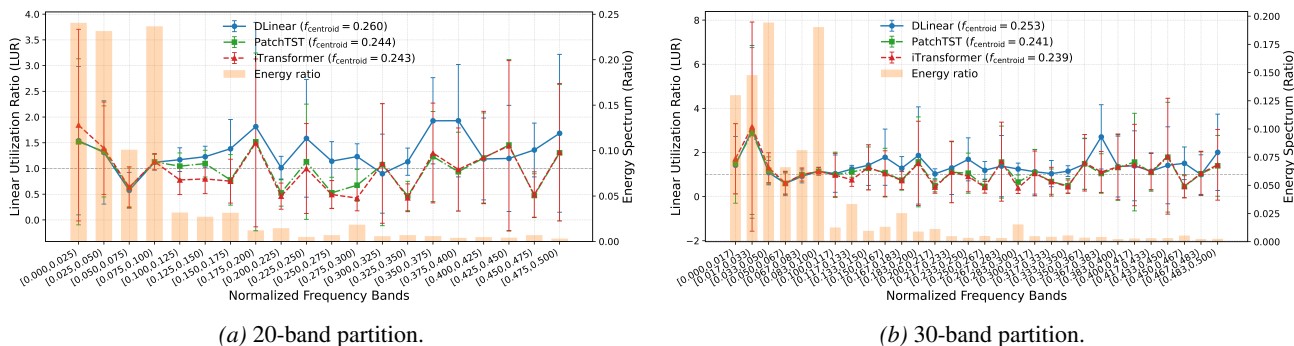

*(a)* 20-band partition.               *(b)* 30-band partition.

*Figure 6.* Band-wise normalized energy and LUR on ETTh1 under different band partitions.

*Table 5.* Multivariate SCP and MSE lower bound versus the number of observed input dimensions $m$ in the synthetic sinusoid mixture experiment.

| $m$ | $\mathcal{P}_{\text{lin}}^{\text{multi}}$ (mean $\pm$ std) | $\text{MSE}_{\text{lb}}^{\text{multi}}$ (mean $\pm$ std) |
|---|---|---|
| 1 | $0.117 \pm 0.121$ | $0.486 \pm 0.067$ |
| 2 | $0.208 \pm 0.145$ | $0.436 \pm 0.080$ |
| 3 | $0.360 \pm 0.130$ | $0.353 \pm 0.072$ |
| 4 | $0.523 \pm 0.169$ | $0.263 \pm 0.094$ |
| 5 | $0.662 \pm 0.151$ | $0.186 \pm 0.083$ |
| 6 | $0.848 \pm 0.030$ | $0.084 \pm 0.017$ |

## C.2. Multivariate Predictability

To validate that our metric captures multivariate predictability, we construct a controlled synthetic example. The input is a $d_x$-dimensional process $\boldsymbol{x}(n) \in \mathbb{R}^{d_x}$ and the target is scalar ($d_y = 1$). We set $d_x = 6$, $d_y = 1$, sequence length $N = 1024$, and number of independent sequences $N_{\text{samples}} = 640$.

For each input dimension $i \in \{1, \ldots, d_x\}$ we generate a sinusoid $x_i(n) = \sin\big(2\pi f_i n + \phi_i\big)$ for $n = 0, \ldots, N-1$, with distinct frequencies $f_i = k_i/L_w$ for a Welch window length $L_w = 128$ and $(k_1, \ldots, k_6) = (3, 5, 7, 9, 11, 13)$, aligned to discrete Fourier bins. The phases are drawn independently as $\phi_i \sim \text{Unif}[0, 2\pi)$ for each $i$ and each sequence. The target signal is defined as a noisy sum of all input components, $y(n) = \sum_{i=1}^{d_x} x_i(n) + \epsilon(n)$, with $\epsilon(n) \sim \mathcal{N}(0, 0.05)$, so that most of the target energy is linearly generated by the $d_x$ inputs.

For each $m \in \{1, \ldots, d_x\}$ we only reveal the first $m$ input dimensions $(x_1, \ldots, x_m)$ and compute the resulting multivariate SCP $\mathcal{P}_{\text{lin}}^{\text{multi}}(m)$ and multivariate MSE lower bound $\text{MSE}_{\text{lb}}^{\text{multi}}(m)$. Both quantities are averaged over the $N_{\text{samples}}$ sequences, and we also report their empirical standard deviations. The numerical results are summarized in Table 5, and the corresponding curves are shown in Fig. 7.

The results exhibit a clear, approximately monotonic trend. As the number of observed input dimensions $m$ increases, the multivariate SCP $\mathcal{P}_{\text{lin}}^{\text{multi}}(m)$ rises almost linearly, while the multivariate $\text{MSE}_{\text{lb}}^{\text{multi}}(m)$ decreases accordingly. As $m$ approaches $d_x$, $\mathcal{P}_{\text{lin}}^{\text{multi}}(m)$ approaches the ideal predictability implied by the signal-to-noise ratio (but does not reach 1 due to spectral estimation error and the injected noise), and $\text{MSE}_{\text{lb}}^{\text{multi}}(m)$ correspondingly approaches zero. Taken together, the table and figure confirm that multivariate SCP faithfully tracks the gain in predictability contributed by additional informative input dimensions.

## C.3. Additional Dataset Evaluation

Table 6 reports detailed long-horizon multivariate forecasting results on the Traffic and Illness datasets for five representative architectures under a matched-information protocol: the history length equals the prediction horizon ($N \in \{96, 192, 336, 720\}$ for Traffic and $N \in \{60, 72\}$ for ILI), with identical preprocessing and no drop-last. We report MSE, MAE, normalized MSE (NMSE), and correlation coefficient $R$, together with the linear MSE lower bound $\text{MSE}_{\text{lb}}$ and SCP-based predictability $\mathcal{P}$ for each task.

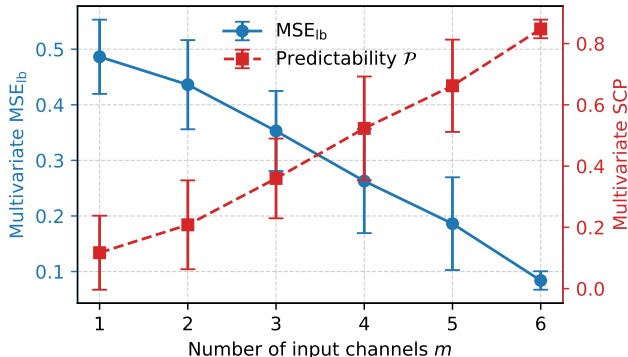

*Figure 7.* Multivariate SCP and MSE lower bound versus the number of input dimensions $m$ in the synthetic sinusoid mixture experiment. Error bars indicate standard deviation across $N_{\text{samples}}$ independent sequences.

*Table 6.* Long-term multivariate forecasting results on Traffic and ILI datasets. We report MSE, MAE, NMSE, and R. **Bold** marks the best (lowest MSE/MAE) per column across models. *Average* rows give the column-wise mean across models. Predictability reports the per-task linear MSE lower bound (MSE$_{\text{lb}}$) and SCP $\mathcal{P}$ (higher is easier).

| Models | Metric | Traffic | | | | ILI | |
|---|---|---|---|---|---|---|---|
| | | 96 | 192 | 336 | 720 | 60 | 72 |
| iTransformer | MSE | **0.394** | **0.385** | **0.388** | **0.416** | 2.001 | 2.186 |
| (Liu et al., 2024) | MAE | **0.269** | 0.269 | 0.274 | 0.290 | 0.954 | 1.033 |
| | NMSE | 0.303 | 0.302 | 0.265 | 0.273 | 0.806 | 1.032 |
| | R | 0.849 | 0.917 | 0.959 | 0.968 | 0.687 | 0.847 |
| TimeMixer | MSE | 0.485 | 0.423 | 0.407 | 0.437 | 2.272 | **1.928** |
| (Wang et al., 2024) | MAE | 0.319 | 0.285 | 0.275 | 0.297 | 0.977 | **0.938** |
| | NMSE | 0.376 | 0.327 | 0.275 | 0.292 | 1.055 | 0.714 |
| | R | 0.919 | 0.939 | 0.971 | 0.960 | 0.783 | 0.781 |
| DLinear | MSE | 0.649 | 0.459 | 0.436 | 0.450 | 2.671 | 2.661 |
| (Zeng et al., 2023) | MAE | 0.396 | 0.305 | 0.296 | 0.306 | 1.083 | 1.114 |
| | NMSE | 0.541 | 0.370 | 0.305 | 0.302 | 1.077 | 1.099 |
| | R | 0.899 | 0.929 | 0.970 | 0.968 | 0.850 | 0.919 |
| PatchTST | MSE | 0.451 | 0.402 | 0.401 | 0.434 | **1.758** | 2.010 |
| (Nie et al., 2023) | MAE | 0.288 | **0.263** | **0.267** | **0.289** | **0.863** | 0.948 |
| | NMSE | 0.356 | 0.321 | 0.277 | 0.288 | 0.753 | 0.792 |
| | R | 0.893 | 0.920 | 0.966 | 0.965 | 0.675 | 0.852 |
| TimesNet | MSE | 0.606 | 0.608 | 0.630 | 0.672 | 2.160 | 1.994 |
| (Wu et al., 2023) | MAE | 0.327 | 0.329 | 0.347 | 0.357 | 0.961 | 0.974 |
| | NMSE | 0.370 | 0.376 | 0.331 | 0.335 | 0.866 | 0.697 |
| | R | 0.960 | 0.969 | 0.946 | 0.983 | 0.820 | 0.742 |
| Average | MSE | 0.517 | 0.455 | 0.452 | 0.482 | 2.172 | 2.156 |
| | MAE | 0.320 | 0.290 | 0.292 | 0.308 | 0.968 | 1.001 |
| | NMSE | 0.389 | 0.339 | 0.291 | 0.298 | 0.911 | 0.867 |
| | R | 0.904 | 0.935 | 0.962 | 0.969 | 0.763 | 0.828 |
| **Predictability** | MSE$_{\text{lb}}$ | 0.803 | 0.616 | 0.400 | 0.636 | 2.151 | 2.681 |
| | $\mathcal{P}$ | 0.514 | 0.619 | 0.760 | 0.610 | 0.560 | 0.466 |

On Traffic, iTransformer consistently achieves the lowest MSE across all horizons. On Illness, TimeMixer and PatchTST achieve better accuracy than the other baselines. Across both datasets, the SCP and linear MSE lower bound remain well aligned with the empirical results, indicating that our predictability-aware metrics continue to agree with, and help interpret, standard error–based evaluations.

### C.4. Comparison with Entropy-based Predictability Metrics

We compare $MSE_{lb}$ with three entropy-based predictability metrics discussed in the related work: permutation entropy (PE), weighted permutation entropy (WPE), and sample entropy (SampEn). Since these metrics are not directly aligned with the MSE forecasting objective, we evaluate them by measuring their correlations with DLinear forecasting errors on ECL.

As shown in Table 7, entropy-based metrics show weak correlations with forecasting errors, whereas $MSE_{lb}$ consistently

*Table 7.* Correlation between predictability metrics and DLinear forecasting error on ECL.

| Metric | 96 | 192 | 336 | 720 |
|---|---|---|---|---|
| PE | 0.086 | 0.010 | 0.109 | 0.117 |
| WPE | 0.168 | 0.189 | 0.217 | 0.252 |
| SampEn | -0.008 | -0.007 | -0.008 | 0.010 |
| $MSE_{lb}$ | 0.880 | 0.867 | 0.909 | 0.864 |

achieves much stronger correlations across all prediction lengths. This indicates that $MSE_{lb}$ is better aligned with instance-level forecasting difficulty under the MSE objective.

### C.5. Long Lookback Window Evaluation

To evaluate the scalability of SCP under long-lookback settings, we conduct an additional experiment on ETTm1 with the prediction length fixed to 336 and history lengths selected from $\{336, 720, 1024, 2048\}$. We report the forecasting MSE, the estimated SCP, their correlation $R$, and the wall-clock time for computing SCP.

*Table 8.* SCP evaluation with different lookback lengths on ETTm1. The prediction length is fixed to 336. Runtime is measured on a Platinum 8358P CPU @ 2.60GHz.

| History Length | 336 | 720 | 1024 | 2048 |
|---|---|---|---|---|
| MSE | 0.371 | 0.384 | 0.366 | 0.360 |
| SCP | 0.268 | 0.246 | 0.230 | 0.276 |
| $R$ | 0.820 | 0.821 | 0.800 | 0.805 |
| Time (ms) | 0.121 | 0.223 | 0.298 | 0.556 |

As shown in Table 8, SCP remains well aligned with realized forecasting error under long lookback windows, with $R$ consistently above $0.8$ even when the history length reaches $2048$. Meanwhile, the computation time increases moderately with sequence length and remains below $1$ ms, indicating that SCP is practical for long-history forecasting benchmarks.

### C.6. Evaluation on a Pretrained Time-series Model

We further examine whether the proposed SCP/$MSE_{lb}$ remains informative for pretrained time-series models with zero-shot forecasting ability. A full study of foundation models involves additional factors, such as cross-dataset transfer and the overlap between pretraining data and downstream predictability regimes, which is beyond the scope of this paper. As an initial evaluation, we test TimerXL with history length fixed to 96 under different prediction lengths (Liu et al., 2025).

*Table 9.* Evaluation of TimerXL under different prediction lengths. The history length is fixed to 96.

| Pred Len | MSE | MAE | NMSE | MSE$_{lb}$ | R |
|---|---|---|---|---|---|
| 96 | 0.2261 | 0.3569 | 1.7246 | 0.2272 | 0.8972 |
| 192 | 0.2891 | 0.4108 | 1.5713 | 0.2637 | 0.9228 |
| 336 | 0.3197 | 0.4391 | 1.4200 | 0.2881 | 0.9148 |
| 720 | 0.4038 | 0.5022 | 1.5398 | 0.3692 | 0.9341 |

As shown in Table 9, $MSE_{lb}$ remains strongly correlated with the realized forecasting error across all prediction lengths, with $R$ ranging from $0.8972$ to $0.9341$. These results suggest that SCP/$MSE_{lb}$ is still effective as an instance-level predictability diagnostic for pretrained forecasting models. We leave a more systematic evaluation of foundation time-series models to future work.

### C.7. Synthetic Nonlinear Predictability Study

We further conduct a controlled synthetic experiment to examine how SCP behaves when the underlying predictability is nonlinear. We generate a raw signal $x(t) = \cos(\omega t + \phi)$ with $\omega = 5\pi/64$, and construct nonlinear components

$T_2(x) = 2x^2 - 1$ and $T_4(x) = 8x^4 - 8x^2 + 1$. The target is defined as

$$y(t) = 0.5x(t) + 1.0T_2(x(t)) + 0.8T_4(x(t)) + \sigma_b\varepsilon(t),$$

where $\varepsilon(t) \sim \mathcal{N}(0, 1)$ and $\sigma_b \sim \log\text{-Uniform}(0.03, 0.20)$.

We evaluate SCP in three representation spaces: the original space $[x]$, an insufficient nonlinear space $[x, T_2(x)]$, and a sufficient nonlinear space $[x, T_2(x), T_4(x)]$.

*Table 10.* Synthetic nonlinear predictability study.

| **Space** | $\mathcal{P}$ | **MSE$_{\text{lb}}$** | **MSE$_{\text{model}}$** | **R** |
|---|---|---|---|---|
| $[x]$ | 0.1312 | 0.8286 | 0.8289 | 0.7894 |
| $[x, T_2(x)]$ | 0.6556 | 0.3285 | 0.3291 | 0.8921 |
| $[x, T_2(x), T_4(x)]$ | 0.9911 | 0.0086 | 0.0088 | 0.9961 |

As shown in Table 10, the original-space SCP is low when the predictable structure is not linearly accessible. After adding nonlinear features, SCP increases substantially and the lower-bound error becomes much closer to the realized model error. In the sufficient feature space $[x, T_2(x), T_4(x)]$, SCP nearly recovers the underlying predictable structure, achieving $\mathcal{P} = 0.9911$ and $R = 0.9961$. This verifies that the nonlinear extension provides a more faithful estimate of predictability when the dominant structure is nonlinear.

