# OpenReview forum: "Beyond Model Ranking: Predictability-Aligned Evaluation for Time Series Forecasting"
_ICML.cc/2026/Conference — ICML 2026 regular_

### Official Review · Reviewer_zCR4 · 2026-03-02

**Soundness:** 2
**Presentation:** 3
**Significance:** 2
**Originality:** 3
**Overall Recommendation:** 4
**Confidence:** 4

**Summary:**

This paper argues that standard aggregate forecasting metrics (e.g., MSE/MAE) conflate model capability with instance difficulty, and proposes a “predictability-aligned” evaluation framework to diagnose model–data mismatch rather than ranking models alone. So the key idea is to evaluate how much of the future segment can be linearly predictable from the history in the frequency domain (per test instance).

**Compliance With Llm Reviewing Policy:**

Affirmed.

**Final Justification:**

I keep my recommendation at borderline accept (4). The paper has some mathematical issues (e.g., the value bound and the LUR interpretation), but these do not seem to affect the main contribution. The authors addressed most of my concerns in the rebuttal. Since I cannot verify the final revised version, I remain cautious and keep the score as borderline accept.

**Key Questions For Authors:**

1. The text says the mean-shift term is “optionally” added, but in algorithm computing $\Delta^2$ seems to be a standard step and includes it in $MSE_{lb}$​ unconditionally. It is unclear to me whether the mean-shift is truly optional.
2. It is widely observed that in time series forecasting, the most informative dependencies for the early future steps are typically near the boundary, is there any specified lag or offset to handle between history and future? It seems from the algorithm that you just feed two vectors (length of N) into Welch CPSD directly.
3. Since LUR is derived from squared coherence, it is strictly scale-invariant. So a model that perfectly captures the frequency phase of $y$ but outputs miscalibrated values by any large factor (e.g., 10k) will have an awful MSE but a perfect LUR score. If this is the case, I suggest the author explicitly acknowledge this limitation and underscore that LUR should be used alongside with MSE, not as a standalone diagnostic.

** If any of these points reflect a misunderstanding on my part, clarification is welcome**

**Limitations:**

yes

**Strengths And Weaknesses:**

Strength:
1. The overall presentation is good and the paper is well-motivated. I totally agree with the critique of the “leaderboard chasing” and shifting the focus toward a instance-level, predictability-aligned diagnostic evaluation is an impactful direction for the time-series community.

2. The SCP is presented as an $O(NlogN)$ per-sample method using Welch method, which is technically sound, plausible and also suitable for large-scale forecasting benchmarks.

3. The proposed LUR metric provides good interpretability and analyzing how different architectures allocate their predictive capacity across frequency bands (i.e., Figure 4) is interesting and offers insights into model behavior that aggregate metrics completely obscure.

Weakness:
1. **(Major) No support on the predictability boundedness.** The paper claims predictability $P_{xy} \in [0,1]$, but the $Var(y)$ is computed from mean-removed $y$ via $\Sigma_f S_{yy}$, while the numerator $MSE$ includes $\Delta^2$ which seems to me that it can exceed $Var(y)$, and yielding a negative $P_{xy}$. It seems there is no proof of the boundedness of the predictability and asserted the [0,1] bound.
2. **(Major) LUR>1 interpretation.** So the paper interprets LUR as “LUR > 1 reveals predictive gains from non-linear or global modeling capabilities”, but the experimental setting is explicitly multivariate forecasting and most of the diagnostic plots/discussion are per-channel. If SCP/LUR are computed “univariately” per channel, then a multivariate model can exceed the “univariate linear information” simply by exploiting cross-channel linear correlations (no nonlinearity required). The paper conflates "beyond this particular univariate linear baseline" with "non-linear modeling capability," which can be a potential logical gap.
3. **The bound claim is not justified with $\Delta$.** A linear predictor can generally fit offsets/trends (depending on the type of predictor). Adding a fixed penalty based on sample-mean mismatch can exceed the achievable MSE of a reasonable linear predictor on trending data, so I think it cannot be a universal lower bound unless the predictor class is very narrowly and specifically constrained.

---

> ### Author Rebuttal · Authors · 2026-03-29
>
> We thank the reviewer for the careful reading and constructive feedback, and especially appreciate the positive assessment of the paper’s motivation, technical soundness, and interpretability. Below, we respond to the concerns in turn and clarify the corresponding revisions we will make in the paper.
>
> **W1: No support on the predictability boundedness.**
>
> **WA1:** Thank you for the careful observation. We therefore revise the definition to $P_{xy}=\max(0,1-\frac{\Delta^2+\sum_f S_e(f)}{Var(\hat y)}) \in[0,1]$,
> so that any negative value is clipped to $0$. This is consistent with our implementation, and we will update the corresponding text to make the boundedness and the role of clipping explicit.
>
> **W2:  LUR>1 interpretation.**
>
> **WA2:** Thank you for this important suggestion. We agree that our current interpretation of $LUR>1$ is too narrow. Since the main diagnostics in the paper are computed per channel using a univariate linear reference, $LUR>1$ can be interpreted as predictive gains beyond this chosen univariate linear baseline. In the multivariate setting, such gains may arise from cross-channel linear correlations, nonlinear dynamics, or global/cross-instance inductive biases. We will revise the manuscript accordingly to avoid attributing $LUR>1$ to nonlinearity alone.
>
> **W3: The bound claim is not justified with $\Delta$.**
>
> **WA3:** Thank you for raising this point.  We will revise the main text to make the scope precise. Specifically, we will clarify that $\sum_{f\in\mathcal{F}} \widehat S_{e}(f)$ yields a lower bound on the MSE of any linear time-invariant predictor after mean removal, which we use as a linear reference error for instance difficulty. For non-stationary data, however, establishing a universal theoretical lower bound for a broad class of predictors is generally difficult. To account for boundary mean mismatch, we therefore add a boundary mean-shift term $\Delta^2 = \big(\mathrm{mean}(\mathbf{y})-\mathrm{mean}(\mathbf{x})\big)^2$, and define $MSE_{\mathrm{lb}} = \Delta^2 + \sum_{f\in\mathcal{F}} \widehat S_{e}(f)$. Accordingly, $\mathrm{MSE}_{lb}$ is a conservative surrogate lower bound, or equivalently a conservative linear reference. The role of $\Delta^2$ is to conservatively account for boundary mean mismatch or local non-stationarity that is not captured by the mean-removed spectral term. If a model can successfully extrapolate offsets or structured trends, this should be viewed as performance beyond the chosen stationary-linear reference. We will revise the manuscript to make this distinction explicit.
>
> **Q1: The mean-shift term is “optionally” added, but in algorithm computing  seems to be a standard step and includes it in ​ unconditionally.**
>
> **QA1:** In the current implementation, experiments, and Algorithm 1, the boundary mean-shift term $\Delta^2$ is included as a standard step in computing $\mathrm{MSE}_{lb}$. We will therefore revise the text to remove the word “optional” and make the definition consistent throughout the paper.
>
> **Q2: It is widely observed that in time series forecasting, the most informative dependencies for the early future steps are typically near the boundary, is there any specified lag or offset to handle between history and future?**
>
> **QA2:** We do not specify a single lag or offset between history and future. Rather, SCP is intended to capture segment-level linear dependence: in the CPSD/coherence formulation, dependence is aggregated over all lags at each frequency, rather than restricted to an explicit lag. In particular, lag or offset information is reflected in the phase of the cross-spectrum, so this formulation can capture phase-shifted or distributed-lag relations that zero-lag time-domain summaries may miss. We will clarify that the coherence is therefore not a zero-lag statistic, but a frequency-domain summary of cross-lag linear dependence, with phase encoding the relative offset between the two segments.
>
> **Q3: Since LUR is derived from squared coherence, it is strictly scale-invariant.**
>
> **QA3:** We appreciate this important clarification request. By construction, LUR is scale-invariant and measures how much predictable structure a model captures. Hence, a model can have high LUR but poor MSE if it captures the correct structural or phase information while mis-scaling the forecast values. This behavior is expected from the definition: LUR measures structural utilization relative to the available linear predictability. We will clarify that SCP/LUR are not intended to replace MSE/MAE, but rather to serve as complementary diagnostics aligned with the squared-error forecasting objective and to provide additional structural insight beyond standard pointwise error metrics.
>
> We thank you again for the constructive feedback, which has helped strengthen our paper. We hope these clarifications satisfactorily address your concerns.

---

> > ### Author Rebuttal · Reviewer_zCR4 · 2026-04-01
> >
> > Thank you for the rebuttal and for taking my feedback into account.
> > I appreciate the corrections to the mathematical formulations, particularly regarding the predictability boundedness and the bound claim. While the revised expressions make the formulation a bit less elegant, they are much more mathematically rigorous and accurate, which is a significant improvement.
> > Since you have successfully addressed the majority of my concerns, I will be maintaining my score for accept.

---

> > > ### Author Response · Authors · 2026-04-02
> > >
> > > Thank you for your follow-up and positive feedback. We sincerely appreciate your acknowledgment that our rebuttal has addressed the main concerns, and we are grateful for your support for accepting the paper. We will incorporate these clarifications and additional explanations into the final version to further improve the paper’s clarity and rigor.

---

### Official Review · Reviewer_f5bF · 2026-03-10

**Soundness:** 3
**Presentation:** 3
**Significance:** 3
**Originality:** 3
**Overall Recommendation:** 5
**Confidence:** 4

**Summary:**

This paper a diagnostic evaluation framework for time-series forecasting. Specifically, it introduces Spectral Coherence Predictability (SCP), an instance-level predictability measure, and Linear Utilization Ratio (LUR), a diagnostic ratio for assessing how effectively a forecasting model exploit linear predictable structure.

**Compliance With Llm Reviewing Policy:**

Affirmed.

**Final Justification:**

The author answered the questions with additional experiments and analysis. I appreciate the insights offered by the manuscript. I decided to raise the score.

**Key Questions For Authors:**

Please see Strength/Weakness

**Limitations:**

The author does not explicitly mention limitations.

**Strengths And Weaknesses:**

# Strengths

S1: The work is well motivated. I agree that current usage of a single aggregate metric is insufficient to characterize modern forecasting performance. To deploy a forecasting model in practice, a closer look at model's strength and weakness is needed, so this work fills the gap.

S2: The empirical study is solid overall. The paper shows generally high correlation between model error and the proposed linear lower bound, and Section 5 provides several insightful analyses of architecture-dependent behavior under different settings.

# Weaknesses / Questions

WQ1: While the framework is analytical and insightful, I think the paper could go one step further toward practical model selection. The current experiments already shows various model strengths across difficulty regimes and frequency bands, but the paper stops there without moving from these diagnosis into a more actionable rule: given a dataset and a set of candidate models, which model should be preferred for which subset of instances or variables? Even if the authors do not advocate a single aggregate metric, some deterministic guidance for model selection or routing would improve the practical impact of the framework.

WQ2: The empirical study focuses on standard supervised forecasting architectures. Recently, there has been works in pretrained / foundation time-series models with claimed zero-shot forecasting ability such as [1] [2]. It would be interesting to evaluate such models under the proposed SCP/LUR framework.

WQ3: from my understanding SCP is fundamentally a linear predictability measure in the original observation space, I think it would be helpful to see a controlled synthetic study where the data are highly predictable but primarily nonlinear, for example through polynomial or other nonlinear transformations with low noise. I think this can help clarify how the original SCP behave when predictability is present but not linearly accessible, and whether the nonlinear extension in the appendix recovers a more faithful notion of difficulty.

[1]: https://arxiv.org/abs/2510.15821
[2]: https://arxiv.org/abs/2505.14766

---

> ### Author Rebuttal · Authors · 2026-03-29
>
> We thank the reviewer for the thoughtful comments and positive assessment of the paper’s motivation and empirical study. We respond to the questions and concerns below.
>
> **WQ1: Moving from diagnosis into a more actionable rule.**
>
> **WA1:** Thank you for this valuable suggestion. SCP/LUR already suggest a natural predictability-aware selection principle: instances with high SCP and strong linear structure are good candidates for lightweight linear models, whereas instances with lower SCP or limited exploitable linear structure are better handled by more expressive nonlinear models. We will make this actionable interpretation more explicit in the revision, and connect it to future directions such as difficulty-based routing, predictability-aware curriculum learning, and hard-sample mining.
>
> **WQ2: Evaluate pretrained / foundation time-series models with claimed zero-shot forecasting ability.**
>
> **WA2:** Thank you for this valuable suggestion. We agree that evaluating pretrained or foundation time-series models, especially in the zero-shot setting, under the proposed SCP/LUR framework is an important and interesting direction. However, a systematic study of such models, involves additional factors beyond the scope of the current paper, such as cross-dataset transfer and the interaction between pretraining data and downstream predictability regimes.
>
> To provide initial evidence, we additionally evaluated a pretrained model, TimerXL, with history length 96 under different prediction lengths. We find that the correlation between $MSE_{lb}$ and realized forecasting error remains consistently high across all settings. This suggests that $MSE_{lb}$ remains informative even for pretrained forecasting models.
>
> | Pred Len | MSE    | MAE    | NMSE   | $\mathrm{MSE}_{lb}$ | $R$    |
> | -------- | ------ | ------ | ------ | ------------------- | ------ |
> | 96       | 0.2261 | 0.3569 | 1.7246 | 0.2272              | 0.8972 |
> | 192      | 0.2891 | 0.4108 | 1.5713 | 0.2637              | 0.9228 |
> | 336      | 0.3197 | 0.4391 | 1.4200 | 0.2881              | 0.9148 |
> | 720      | 0.4038 | 0.5022 | 1.5398 | 0.3692              | 0.9341 |
>
> We will incorporate this clarification in the revision and highlight pretrained/foundation models as a natural extension of the proposed diagnostic framework.
>
> **WQ3: How the original SCP behave when predictability is present but not linearly accessible.**
>
> **WA3:** We thank the reviewer for this valuable suggestion. The framework can be extended by mapping the raw observations into a higher-dimensional feature space, where nonlinear structure may become linearly accessible, and then applying SCP in that transformed space.
>
> To clarify this point, we added a controlled synthetic experiment. We generate a raw signal as $x(t)=\cos(\omega t+\phi)$, where $\omega=5\pi/64$ and $\phi$ is a random phase. We then construct nonlinear polynomial components $T_2(x)=2x^2-1$ and $T_4(x)=8x^4-8x^2+1$, and define the target as $y(t)=0.5\,x(t)+1.0\,T_2(x(t))+0.8\,T_4(x(t))+\sigma_b\varepsilon(t)$, where $\varepsilon(t)\sim\mathcal{N}(0,1)$ and the per-instance noise scale is sampled as $\sigma_b\sim\log\text{-Uniform}(0.03,0.20)$. Each instance has length $512$, we use $128$ instances per trial, and evaluate both the SCP-based linear lower bound and the corresponding linear predictor.
>
> We consider three representation spaces: the original space $[x]$, an insufficient transformed space $[x,T_2(x)]$, and a sufficient transformed space $[x,T_2(x),T_4(x)]$. The resulting statistics are summarized below.
>
> | Space               | $\mathcal{P}$ | $MSE_{lb}$ | $MSE_{model}$ |    $R$ |
> | - | ------------: | ---------: | ------------: | -----: |
> | $[x]$               |        0.1312 |     0.8286 |        0.8289 | 0.7894 |
> | $[x,T_2(x)]$        |        0.6556 |     0.3285 |        0.3291 | 0.8921 |
> | $[x,T_2(x),T_4(x)]$ |        0.9911 |     0.0086 |        0.0088 | 0.9961 |
>
> The results show a clear progression as the nonlinear structure becomes more fully represented. In the original observation space, the process is still highly predictable in principle, but most of that predictable structure is not linearly accessible from $x$ alone. Accordingly, the original-space SCP is low, with $\mathcal{P}=0.1312$, and the corresponding linear lower-bound error is high, with $MSE_{lb}=0.8286$. Once the representation becomes sufficient, SCP almost fully recovers the underlying predictable structure, with $\mathcal{P}=0.9911$, $MSE_{lb}=0.0086$, and a much tighter alignment with realized error, reaching $R=0.9961$. This controlled example shows that the nonlinear extension is particularly useful when the dominant predictable structure is nonlinear, providing a more faithful notion of difficulty in such settings.
>
> We thank you again for the constructive feedback, which has helped strengthen our paper. We hope these clarifications satisfactorily address your concerns.

---

> > ### Author Rebuttal · Reviewer_f5bF · 2026-04-03
> >
> > The authors have addressed my concern with supplemental experiment. I will raise my score to accept. Good luck.

---

> > > ### Author Response · Authors · 2026-04-03
> > >
> > > Thank you for your follow-up and positive feedback. We sincerely appreciate your acknowledgment that our rebuttal has addressed the main concerns, and we are grateful for your support for accepting the paper. We will incorporate these clarifications and additional explanations into the final version to further improve the paper’s clarity and rigor.

---

### Official Review · Reviewer_dFYv · 2026-03-10

**Soundness:** 4
**Presentation:** 3
**Significance:** 4
**Originality:** 3
**Overall Recommendation:** 5
**Confidence:** 3

**Summary:**

Current deep neural forecasters are mostly compared under point-wise metrics such as mean squared error (MSE) and mean absolute error (MAE). This approach, however, does not necessarily reflect the true forecasting capability of the model. In the meantime, other explainable sequence metrics may suffer from computational inefficiency or reduced expressivity. To resolve this dilemma, the paper proposes a new metric based on traditional spectral analysis. This can measure the forecasting ability efficiently and help analyse the underlying problems of certain models. The experiments show promising alignment effects and inspiring diagnostics with existing forecasting models.

**Compliance With Llm Reviewing Policy:**

Affirmed.

**Final Justification:**

The overall quality of the paper is sufficiently high, and the proposal may be meaningful to the community. There are some minor concerns about the technique detail; the authors' responses indicated that they can address these concerns in the final revision, therefore, I maintain my positive score.

**Key Questions For Authors:**

Q1: Would the inherent noise and extreme points of data affect the spectral density estimation in LUR, causing some numerical instability?

Q2: Could authors elaborate on the reason why adding boundary mean shift can lead to a tight lower bound on the MSE of a linear time–invariant predictor? It seems that the intuitions of eq.(4~8) are not obvious or less interpretable to the unfamiliar reader.

**Limitations:**

Neither is included.

**Strengths And Weaknesses:**

Strengths:
>	- The manuscript is well-written, the logic is very clear and easy to follow.
>	- The proposed methods are built on mature methods of spectral analysis, which are naturally simple and effective by FFT.
>	- The experiment results seem to show that the proposed metric is well aligned with existing forecasting behaviors, verifying the effectiveness of proposals.
>	- The diagnostic metric, LUR, is quite intuitive and useful. The experimental analyses of the predictability drift are meaningful and perhaps overlooked in the field.

Weaknesses:
>	-  It seems that the proposed method is a direct transfer from classic linear metrics, such as Pearson correlations and the intrinsic predictability via Bayes risk, to their spectral variants. While I appreciate the conceptual cleanliness and technical solidness of the method, **I am not sure about the novelty of this proposal, considering that I am not very familiar with the most related literature.** Also, I think the related work part should be strengthened, which may be too shallow to reflect its position.
> - The authors measure all the coherence in the frequency domain. If we make our analysis or evaluation totally on the frequency domain, are there any risks of losing the temporal correlations, which are also very significant for evaluating the forecasting performance?

---

> ### Author Rebuttal · Authors · 2026-03-29
>
> We thank the reviewer for the thoughtful comments and especially appreciate the positive assessment of the manuscript’s clarity, the technical soundness of the spectral formulation, and the usefulness of the diagnostic analyses. Below, we address the concerns in turn and clarify the revisions we will make.
>
> **W1: The contribution and related work part should be strengthened.**
>
> **WA1:** Thank you for this thoughtful comment. Our contribution is a forecasting-oriented diagnostic framework. Specifically, we (i) define a coherence-based per-instance difficulty reference explicitly aligned with multi-horizon MSE, (ii) connect it to intrinsic predictability through a Bayes-risk formulation as a tractable surrogate, (iii) introduce band-wise and predictability-stratified evaluation protocols to expose architectural biases, and (iv) use this lens to quantify predictability drift and its interaction with model behavior.
>
> We will also clarify the distinction from prior work. Prior predictability studies have mostly focused on upper bounds or intrinsic predictability itself, rather than linking predictability to realized model behavior through a diagnostic evaluation framework. Moreover, they are not directly aligned with the real-valued multi-step forecasting setting. Prior spectral forecasting work also has a different goal: it mainly uses frequency-domain structure to build stronger forecasting models, whereas we use spectral coherence to analyze predictability. We will revise the Related Work section to make this positioning explicit.
>
> Specifically, we will revise the Related Work section to state more explicitly: *Existing predictability studies have primarily focused on intrinsic predictability or theoretical performance limits [1,2,3], but are not directly designed for real-valued multi-step time-series forecasting, and have rarely been framed as a predictability-centered evaluation framework for understanding realized forecasting performance, model bias across regimes, and temporal predictability drift.*
>
> **W2: All the coherence in the frequency domain.**
>
> **WA2:** We appreciate this important clarification request. Under the Wiener–Khinchin theorem, the spectrum and the autocorrelation function are equivalent second-order representations, so SCP/LUR preserve temporal dependence in the frequency domain rather than discard it. In addition, this spectral formulation is computationally efficient, which makes it practical as an instance-level diagnostic for large-scale forecasting benchmarks.
>
> **Q1: Noise and extreme points of data cause numerical instability**
>
> **QA1:** Thank you for raising this point. Our framework mitigates this through Welch’s method, which uses windowing and segment-wise averaging to stabilize the predictability estimates. Inherent noise mainly manifests as lower coherence and larger unexplained variance, rather than numerical explosion or spurious predictability. Since such noisy instances are also harder for the forecasting model itself, the resulting lower predictability is aligned with higher realized error rather than being an artifact of the estimator. Empirically, Fig. 3 already reflects this regime.
>
> **Q2: A tight lower bound on the MSE of a linear time–invariant predictor**
>
> **QA2:** We appreciate this important clarification request. The key idea is that SCP provides a computable surrogate for intrinsic predictability under MSE. In Eq. (4), squared coherence gives the explained fraction of target power at each frequency, while Eq. (5) defines the unexplained residual power. Summing this residual spectrum, $\sum_{f\in\mathcal{F}} \widehat S_{e}(f)$, yields the minimum achievable error of the optimal linear time-invariant predictor after mean removal, following the standard Wiener filtering and orthogonality principle[4]. Hence, it provides a lower bound on the MSE of any linear time-invariant predictor after mean removal. The $\Delta^2$ captures mismatch in the mean component between history and future, for example due to local mean shift or non-stationary trend behavior, which is not reflected in the mean-removed spectral term. Therefore, $\mathrm{MSE}_{lb}$ should be interpreted as a conservative surrogate lower bound, or equivalently a conservative linear reference. We will clarify this distinction more explicitly in the manuscript and add a short derivation in the appendix.
>
> We thank the reviewer again for the careful reading and constructive suggestions. We hope these clarifications satisfactorily address your concerns.
>
> [1] Limits of predictability in human mobility. *Science*, 2010
>
> [2] Contrasting social and non-social sources of predictability in human mobility. *Nature Communications*, 2022
>
> [3] Quantifying and estimating the predictability upper bound of univariate numeric time series. SIGKDD, 2024
>
> [4] An Introduction to the Theory of Random Signals and Noise. 1985

---

> > ### Author Rebuttal · Reviewer_dFYv · 2026-04-01
> >
> > I appreciate the authors' clarifications. Please add these additional explanations to the final version to help with a deep understanding. I decide to maintain my score, good job.

---

> > > ### Author Response · Authors · 2026-04-02
> > >
> > > Thank you for your follow-up and positive feedback. We sincerely appreciate your acknowledgment that our rebuttal has addressed the main concerns, and we are grateful for your support for accepting the paper. We will incorporate these clarifications and additional explanations into the final version to further improve the paper’s clarity and rigor.

---

### Official Review · Reviewer_BuHa · 2026-03-12

**Soundness:** 2
**Presentation:** 3
**Significance:** 2
**Originality:** 3
**Overall Recommendation:** 4
**Confidence:** 3

**Summary:**

This paper introduces two diagnostic metrics for time series forecasting evaluation, enabling fairer model comparisons by accounting for task difficulty and more fined model behaviors. Standard metrics like MSE conflate model capability with the inherent difficulty of the data, obscuring meaningful comparison. Therefore, the authors propose Spectral Coherence Predictability (SCP) as the predictability metric, which leverages frequency-domain coherence between historical and future segments, and Linear Utilization Ratio (LUR) as the frequency-resolved diagnostic metric that measures linear predictability across frequency bands. Experiments on synthetic and real-world benchmarks demonstrate that SCP correlates with actual forecasting errors across multiple architectures, and LUR can reveal model performance across different frequency bands.

**Compliance With Llm Reviewing Policy:**

Affirmed.

**Key Questions For Authors:**

1. Does the $\Delta^2$ term risk over-compensating for predictable trend shifts ? By treating the mean mismatch as unpredictable difficulty into the lower bound of MSE (eq. 6), the framework might undervalue models that possess strong trend modeling capabilities.
2. For short-term time series, would the low spectral resolution lead to unreliable predictability estimates? Specifically, how does SCP maintain its robustness when the window length is insufficient to accurately resolve low-frequency components?

**Limitations:**

1. Empirical Limitations: The evaluation is conducted on a limited set of datasets, metrics, and time series lengths, which is insufficient to fully verify the generalization of the proposed metrics (see weakness section).
2. Method Limitations: The method provides no uncertainty estimates for predictability scores and may be not reliable enough for short time series (see key questions).

**Strengths And Weaknesses:**

Strengths:

1.The research on predictability and diagnostic metrics offers a valuable perspective for time series forecasting evaluation, moving beyond raw prediction error as the sole criterion.

2.The experiment not only demonstrates the effectiveness of the proposed index, but also provides insightful analysis of time-varying predictability.

Weaknesses:
1. The evaluation was conducted on datasets with clear seasonal patterns and dominant frequencies. How do SCP and LUR generalize to time series datasets with weak or irregular spectral structures ?
2. The experiments lack comparisons with other predictability metrics discussed in the related work section.
3. The evaluation is limited to deterministic metrics such as MAE, MSE, and NSE. How does SCP correlate with probabilistic metrics like CRPS?
4. Many existing benchmarks adopt arbitrary or long lookback windows, like 2048. How does the wall-lock time overhead on calculating SCP scale with increasing time series length ? And does the estimated predictability remain well-aligned with MSE across the datasets that lack a primary frequency, particularly when longer historical sequences are used ? Section 5.7 just provides a limited evaluation in ETTh1.
5. Could the authors provide confidence intervals or uncertainty estimates for SCP and LUR ?

---

> ### Author Rebuttal · Authors · 2026-03-30
>
> We thank the reviewer for the careful reading and constructive feedback, and appreciate the recognition of the paper’s motivation and diagnostic value. Our responses are below.
>
> **W1: With weak or irregular spectral structures.**
>
> **WA1:** SCP/LUR do not assume a single dominant frequency or strong periodicity. Both aggregate history–future coherence over the full frequency grid, so they remain well-defined for series with weak or irregular spectral structure. In such cases, SCP is naturally lower, while the model MSE tends to be higher because such structure is harder to exploit reliably. Empirically, Fig. 3 already reflects this regime.
>
> **W2: Comparisons with other predictability metrics discussed in the related work section.**
>
> **WA2:** Prior predictability measures do not naturally yield an instance-level MSE-aligned reference. However, we further evaluated permutation entropy (PE), weighted permutation entropy (WPE), and sample entropy (SampEn) on ECL, and compared their **correlations** with DLinear forecasting error against our proposed $MSE_{lb}$:
>
> |            | 96    | 192   | 336   | 720   |
> | - | -| - | - | - |
> | PE         | 0.086 | 0.010 | 0.109 | 0.117 |
> | WPE        | 0.168 | 0.189 | 0.217 | 0.252 |
> | SampEn     | -0.008 | -0.007 | -0.008 | 0.010 |
> | $MSE_{lb}$ | 0.880 | 0.867 | 0.909 | 0.864 |
>
> Across all prediction lengths, $MSE_{lb}$ is much more strongly aligned with realized forecasting difficulty.
>
> **W3: Correlate with probabilistic metrics like CRPS?**
>
> **WA3:** Our theory starts from Bayes risk under MSE, and SCP is constructed as an MSE-aligned surrogate difficulty reference in this setting. Extending the framework to probabilistic forecasting is an important direction for future work, for example by redefining predictability through the Bayes risk of a proper scoring rule such as CRPS under additional assumptions.
>
> **W4: Experiment with long lookback windows.**
>
> **WA4:** To address long-lookback settings, we additionally evaluated DLinear on ETTm1 with prediction length fixed at $336$ and history lengths $\{336,720,1024,2048\}$:
>
> |           | 336   | 720   | 1024  | 2048  |
> | - | - | -| - | -|
> | MSE       | 0.371 | 0.384 | 0.366 | 0.360 |
> | SCP       | 0.268 | 0.246 | 0.230 | 0.276 |
> | $R$       | 0.820 | 0.821 | 0.800 | 0.805 |
> | Time (ms) | 0.121 | 0.223 | 0.298 | 0.556 |
>
> The results show that the correlation between SCP and realized forecasting error remains consistently above 0.8 even when the lookback window is extended to 2048. At the same time, the computation time stays **below 1 ms** on a Platinum 8358P CPU @ 2.60GHz, supporting the practical scalability of the method.
>
> **W5: Provide confidence intervals for SCP and LUR.**
>
> **WA5:** Yes. Confidence intervals for SCP and LUR are reported in Table 2, Table 5, Fig. 6, and Fig. 7. These results support that both quantities are stable under reasonable estimation choices. At the same time, predictability and realized model error vary substantially across channels, instances, and frequency bands, so Table 1 reports mean statistics. As one example, for DLinear on Weather, the variation in $MSE_{lb}$ closely tracks model MSE, explaining the consistently high correlation:
>
> |            | 96                | 192               | 336               | 720               |
> | -- | --| ---- | - | -- |
> | MSE        | 0.197 $\pm$ 0.146 | 0.225 $\pm$ 0.151 | 0.263 $\pm$ 0.172 | 0.315 $\pm$ 0.167 |
> | $MSE_{lb}$ | 0.186 $\pm$ 0.134 | 0.244 $\pm$ 0.160 | 0.278 $\pm$ 0.209 | 0.317 $\pm$ 0.168 |
> | $R$        | 0.924             | 0.931             | 0.911             | 0.923             |
>
> **Q1: Does the $\Delta^2$ term risk over-compensating for predictable trend shifts?**
>
> **QA1:**  $\sum_{f\in\mathcal{F}} \widehat S_e(f)$ captures the lower-bound component of local stationary linear structure. For non-stationary series, a precise universal bound is generally unavailable, so $\Delta^2$ is included only as a boundary mean-shift correction. Thus, $MSE_{lb}$ should be interpreted as a conservative stationary-linear reference. Empirically, Table 1 and Figs. 2-3 show that SCP/$MSE_{lb}$ remains highly correlated with realized model error even for nonlinear models and non-stationary data.
>
> **Q2: Short-term time series.**
>
> **QA2:** Short history windows reduce spectral resolution, especially at low frequencies. In such cases, unresolved low-frequency structure is conservatively absorbed into the residual spectrum, leading to lower SCP and higher $MSE_{lb}$. But, the same limitation also applies to the forecasting model itself. For practically reasonable history lengths, however, the correlation between $MSE_{lb}$ and per-sample model errors remains consistently high, as shown in Table 3, typically with $R \ge 0.80$.
>
> We thank you again for the constructive feedback, which has helped strengthen our paper. We hope these clarifications satisfactorily address your concerns.

---

> > ### Author Rebuttal · Reviewer_BuHa · 2026-04-03
> >
> > Thank you for the helpful rebuttal. My concerns have been addressed well, and the additional results helped clarify the paper and strengthen its overall quality.
> >
> > Since I have already given a positive score, I will keep my score.

---

> > > ### Author Response · Authors · 2026-04-05
> > >
> > > Thank you for your follow-up and positive feedback. We sincerely appreciate your acknowledgment that our rebuttal has addressed the main concerns, and we are grateful for your support for accepting the paper. We will incorporate these clarifications and additional explanations into the final version to further improve the paper’s clarity and rigor.

---

### Decision · Program_Chairs · 2026-04-30

**Decision:**

Accept (regular)

**Comment:**

This paper introduces two new metrics to assess time series forecasting models to deconflate model predictive capabilities from instance difficulty. The first metric - spectral coherence predictability - measures the linearity of the data in the spectral domain while the second metric - linear utilization ratio - measures the degree that a model relies on this linearity in its predictions. Overall, this paper provides useful insight and well-grounded metrics that provide insight beyond standard metrics of MSE/MAE. This contribution has high significance in that it can counteract “leaderboard chasing” in time-series model development, and the experimental section provides compelling evidence and detailed insight into the metrics. During the rebuttal period, the authors sufficiently addressed issues regarding contextualization experiments, some technical mathematical issues, and more experimental comparisons. Overall, I believe this work will be a valuable contribution to ICML and I recommend its acceptance.